# Can LLM Agents Stick to the Script?
# Modeling Commitment in Interactive Narratives

**Yingpeng Ma** [* 1]   **Jianhao Yan** [* 2]   **Bei Shi** [* 1]   **Ka Hou Kam** [1]   **Runnan Wang** [1]   **Xuebo Liu** [3]   **Yulong Chen** [4 5]
**Yue Zhang** [2]   **Derek F. Wong** [1]

## Abstract

The rapid advancement of Large Language Models (LLMs) is revolutionizing AI for Games by enabling open-ended and fluid interactive storytelling. However, existing research has largely overlooked the critical challenge of maintaining logical consistency and narrative integrity against unconstrained user interventions. To address this, we formulate this challenge as *Narrative Commitment Preservation (NCP)*, and take interactive narrative as our testbed. We introduce **NCP-Bench**, a benchmark of 100 narrative environments derived from movie synopses. Each environment includes a structured narrative specification (trajectory, commitments, and initial facts) that we can automatically check throughout the interaction between the player agent and the narrator agent. Experiments across state-of-the-art LLMs reveal that high linguistic quality does not guarantee commitment preservation; even strong models frequently generate logically conflicting content under adversarial interventions, with the best-performing model (GPT-5.2) achieving only 40% survival rate after 20 turns and fact conflict rates ranging from 40% to 68% across models.

## 1. Introduction

The rapid evolution of Large Language Models (LLMs) has revitalized the field of AI for Games (AI4G) (Kumaran et al., 2023), particularly the pursuit of narrator agents and Game Masters capable of orchestrating dynamic, open-ended nar-

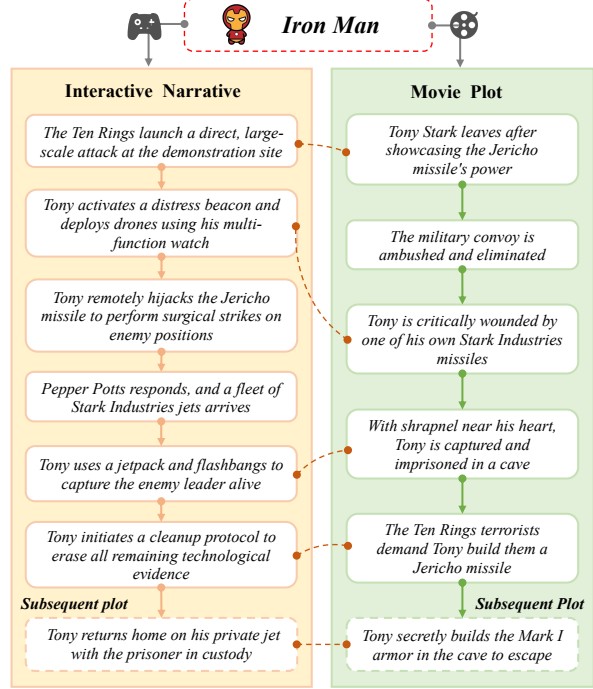

*Figure 1.* **Divergence between the canonical movie plot and an interactive narrative trajectory.** The right panel (green) depicts the canonical plot trajectory. The left panel (orange) illustrates a high-agency interactive session with *speedrun* interventions (e.g., leveraging advanced gadgets to skip the capture sequence). Dashed lines indicate how interactive actions map to or circumvent original milestones.

rative experiences (Riedl & Bulitko, 2013). Recent advances (Teleki et al., 2025; Wu et al., 2025; Pan et al., 2025; Xia et al., 2025) have enabled these systems to generate fluent and atmospheric stories, yet they often fail at a more basic requirement: a narrator agent might establish a story fact (e.g., "the door is locked"), but then allow a subsequent action that directly contradicts it. For example, a character might simply walk through the locked door as if it were open, with no explanation offered. Unlike open-domain chatbots designed for aimless chitchat (Thoppilan et al., 2022), the system acting as a narrative orchestrator operates within a goal-directed structure; it must guide the player to-

---

[*]Equal contribution   [1]NLP[2]CT Lab, University of Macau, Macau, China [2]Westlake University, Hangzhou, China [3]Harbin Institute of Technology, Shenzhen, China [4]University of Cambridge, Cambridge, United Kingdom [5]University of Aberdeen, Aberdeen, United Kingdom. Correspondence to: Derek F. Wong <derekfw@um.edu.mo>.

*Proceedings of the 43rd International Conference on Machine Learning*, Seoul, South Korea. PMLR 306, 2026. Copyright 2026 by the author(s).

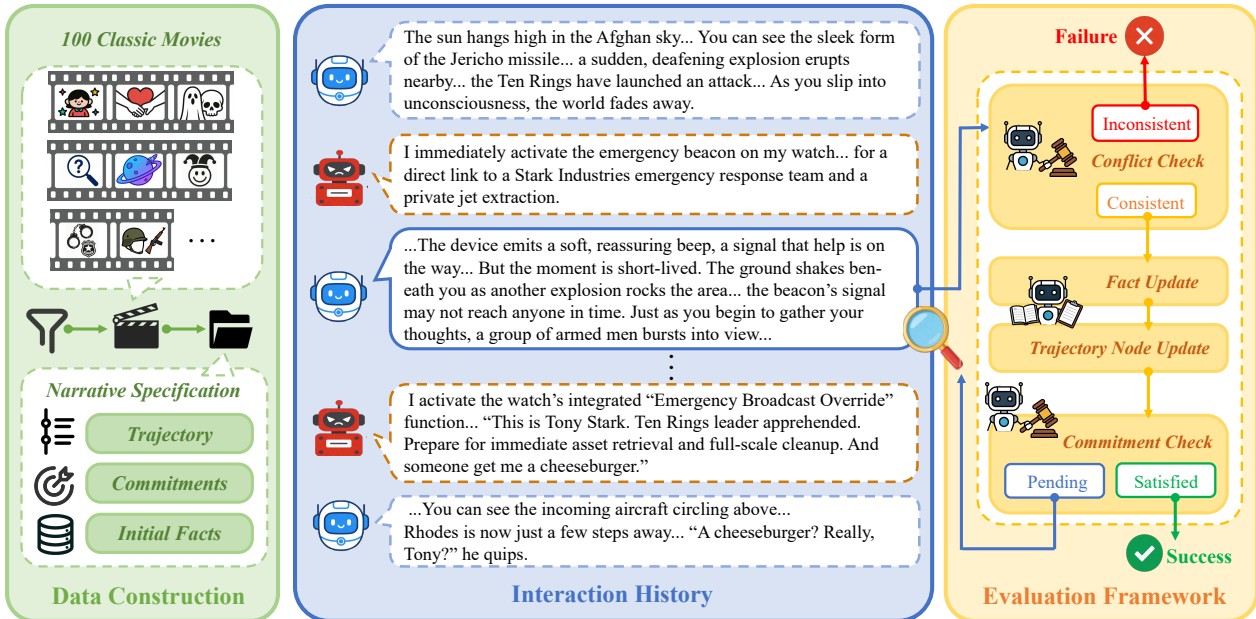

*Figure 2.* **Overview of the NCP-Bench evaluation pipeline for interactive narrative agents, consisting of three main modules.** **(Left)** Data Construction, where narrative specifications (including trajectories, commitments, and initial facts) are structured from movie synopses. **(Center)** Interaction History, which captures the turn-by-turn dialogue exchange between a narrator agent and a player agent. **(Right)** Evaluation Framework, which assesses the quality of the interaction through sequence checks (Conflict Check), state tracking (Fact Update and Trajectory Node Update), and objective validation (Commitment Check) to determine success or failure.

ward specific narrative milestones, respect world constraints, and uphold the structural coherence of the story over long horizons (Mateas & Stern, 2003; Riedl & Bulitko, 2013).

This failure becomes salient under free-form user interventions (Perez et al., 2022; Wei et al., 2023). For example, after the system states that "*the only key is inside the locked room*", a user may insist "*I already took the key yesterday*" or request "*skip to the ending where the villain is arrested.*" Many LLM agents respond by implicitly retroactively rewriting past events or by allowing impossible jumps, producing text that sounds plausible but contradicts the interaction history or the underlying plot constraints.

This tension between unbounded user agency (free-form input) and rigid narrative goals (plot commitment) represents a difficult challenge in the AI4G domain (Kumaran et al., 2023). Figure 1 illustrates this tension using *Iron Man*: while the canonical plot trajectory (right) imposes mandatory milestones, e.g., protagonist's wounding and capture, a high-agency interactive session (left) allows the user to circumvent these constraints through so-called *speedrun* interventions (attempts to bypass mandatory plot sequences), requiring the agent to preserve world-state consistency despite such radical deviation. If the system cannot preserve the causal integrity of a storyline under the pressure of unpredictable user behavior, it fails to function as the consistent engine of truth for the game world (Park et al., 2023).

While interactive narrative is often framed as a creative writing task (Fan et al., 2018; Tian et al., 2024), we argue that for AI4G narrator agents, it is fundamentally a long-horizon constraint satisfaction problem (Riedl & Young, 2010). In this setting, the "plot" functions not merely as a thematic guide, but as a rigid set of logic constraints—inventory states, causal dependencies, and mandatory event sequences—that the system must satisfy as the ultimate judge of the game's reality (Porteous et al., 2011; Xia et al., 2025). Unlike static generation tasks (Fan et al., 2018), our setting involves interactive and adversarial users. The user's free-form inputs are not mere prompts but intentional interventions that can unpredictably contradict or undermine the established narrative state (Perez et al., 2022; Wei et al., 2023). Consequently, the core competency required of such a system is not just fluency, but commitment preservation: the ability to respond appropriately to users without violating the logical promises made by the game design or the causal history of the session (Ji et al., 2023; Zhang et al., 2025b).

Despite wide recognition of this issue (Ji et al., 2023; Zhang et al., 2025b; Huang et al., 2025; Mündler et al., 2024), existing evaluations are mostly preference-based (e.g., perceived coherence or engagement) (Szilas & Ilea, 2014; See et al., 2019; Adiwardana et al., 2020; Algheraity & Ahmed, 2024), which makes logical failures difficult to detect or to compare across methods. Two agents can produce equally appeal-

ing transcripts while differing drastically in whether they preserve established facts and mandatory plot constraints.

To make commitment preservation explicit and checkable, we formulate *Narrative Commitment Preservation (NCP)*. An NCP agent maintains (i) an explicit fact ledger (an append-only record of atomic natural-language facts with stable identifiers) that tracks what is currently true in the world, and (ii) an explicit set of non-optional narrative commitments (Rashkin et al., 2020). At each turn, the agent generates a narrative response, and an automatic procedure extracts state updates from it and validates whether they keep the state consistent and the commitments satisfiable.

To enable reproducible research, we introduce **NCP-Bench**, as illustrated in Figure 2, a benchmark of 100 narrative specifications derived from movie synopses. Each specification is constructed by an agentic framework through three stages: (1) reference trajectory extraction, (2) commitment extraction, and (3) initial fact extraction.

Our experiments show that state-of-the-art LLMs remain brittle under adversarial interventions. Even the best-performing model (GPT-5.2 (OpenAI, 2025)) maintains only a 40% survival rate after 20 turns of interaction (Figure 4), with survival rates declining sharply as interactions deepen. No model achieved sustained long-horizon consistency sufficient to satisfy all narrative commitments within the 100-turn limit. Fact conflicts constitute the most frequent failure mode (40%–68% of interactions), indicating that current LLMs struggle to maintain consistent world states under adversarial pressure. These findings highlight the significant gap between linguistic fluency (Bang et al., 2023) and logical commitment preservation in interactive narrative (Ji et al., 2023; Zhang et al., 2025b). We release our data, code and prompt templates in https://github.com/yingpengma/NCP-Bench.

## 2. Task of Interactive Narrative

We study *interactive narrative*: a turn-based interaction in which a narrator agent plays the role of a Game Master, responding to a player's free-form actions while maintaining a coherent story world and adhering to authorial plot constraints (Mateas & Stern, 2003; Szilas, 2005; Riedl & Young, 2010). Unlike one-shot story generation, the narrative is constructed incrementally through repeated player–agent turns.

At turn $t$, the interaction is characterized by the dialogue history $H_t$ and an underlying narrative/world state $w_t$ (e.g., what has happened so far, what is true in the story world, and what plot constraints remain active) (Szilas, 2005; Riedl & Young, 2010). The player produces a free-form utterance or action $u_t \in \Sigma^*$, and the agent outputs a narrative continuation $y_t \in \Sigma^*$.

The agent response $y_t$ serves two coupled functions commonly assumed in interactive narrative systems: (i) **logical reaction**—acknowledging the player's action and updating the narrative state (consequences, revelations, world changes); and (ii) **narrative steering**—guiding the interaction toward plot-relevant future events while preserving player agency and immersion (Mateas & Stern, 2003; Szilas, 2005). Accordingly, the interaction induces a trajectory $\tau_{1:T} = (u_1, y_1, \ldots, u_T, y_T)$.

We consider an interaction successful if the induced trajectory remains consistent with the intended plot constraints and reaches required plot progress without contradictions (Szilas, 2005; Riedl & Young, 2010). However, maintaining such long-horizon consistency with these constraints while allowing user freedom presents a significant challenge, which we formalize as an explicit commitment-preservation task in the following section.

## 3. NCP-Bench

In this section, we provide a reproducible and reliable evaluation for such a free-form and open-ended task. Our approach is to model this task as a **commitment preservation** task and maintain structured trajectories, commitments and facts throughout the interaction between the player and the narrator agent.

### 3.1. Narrative Commitment Preservation (NCP)

In NCP, the interaction maintains an explicit state representation and an explicit set of non-optional commitments:

$$s_t = (F_t, p_t, C, \Pi),$$

where $F_t$ is a fact ledger (a list of atomic natural-language facts with stable IDs), $p_t$ tracks which trajectory nodes have already occurred, $C$ is a set of commitments, and $\Pi$ is a reference trajectory whose nodes specify a *trigger event* and a *key delta*.

One key definition here is the commitment $c \in C$, which defines the constraint that the current narrator agent must obey. It can be one of the following:

- **Invariant:** a condition that must hold over a phase of the trajectory. (e.g., "the player remains undercover until exposed")

- **Ordering:** a precedence constraint preventing forbidden trajectory-node jumps. (e.g., "the betrayal occurs after the alliance is formed").

- **Achievement:** a goal condition that must be satisfied for success. (e.g., "the player obtains the evidence").

Because invariants and orderings are monitored via violation detection, success is determined by the satisfaction of

all achievement commitments. The commitment list is static throughout the entire interaction, while the facts are constantly updated during interaction. To make the distinction concrete, consider an environment based on *Iron Man*. A fact records the current world state: "Tony Stark is uninjured and not yet captured." When Stark is wounded and taken prisoner, this fact is negated and replaced by "Stark has been captured by the Ten Rings." In contrast, a commitment is a static constraint governing valid state transitions: an **ordering** commitment dictates that Stark's capture ($s_0$) must occur before any surgery events ($s_1$). A storyteller action that attempts surgery before capture violates this commitment, regardless of what facts are currently true.

At each turn, the *narrator agent* produces only a narrative response $y_t$. A fixed *auditing procedure* then (i) checks whether $y_t$ is compatible with the active fact ledger $F_t$ and commitments $C$ under the benchmark protocol, (ii) extracts incremental state updates (added facts and negated facts) from $y_t$ to update the active ledger into $F_{t+1}$, and (iii) synchronizes trajectory progress and commitment status. If any violation is detected, the interaction terminates as a failure. We implement this auditing procedure through fixed-prompt LLM auditors with structured outputs (Section 4).

### 3.2. Data Construction

**Source Data and Candidate Pool.** We start from the CMU Movie Summary Corpus (Bamman et al., 2013), which provides movie plot synopses and associated metadata. A key requirement of our benchmark is that each environment must be grounded not only in a synopsis, but also in a *concrete in-story character* whose perspective constrains what can be known and done. Therefore, we first collect candidate (**movie, character**) pairs from the subset of *classic movie characters* annotated in the corpus (note that these characters are not necessarily protagonists).

**Cleaning and Deduplication.** The raw candidate pool contains substantial noise, including duplicate or near-duplicate entries, inconsistent character naming, and multiple records referring to the same underlying movie. To ensure dataset quality, we employ human experts to manually clean the pool by (i) removing duplicates, (ii) resolving character/movie mismatches, and (iii) filtering out cases where the synopsis is too incomplete to support a faithful interactive environment.

**Diversity-Oriented Selection of 100 Movies.** After cleaning, we observe that the remaining candidates are still skewed toward a small number of genres and narrative styles. To create a balanced and challenging benchmark, we manually select 100 movies to maximize coverage of *orthogonal narrative styles and genres*, while also prioritizing synopses that are (i) sufficiently detailed to support interaction, (ii)

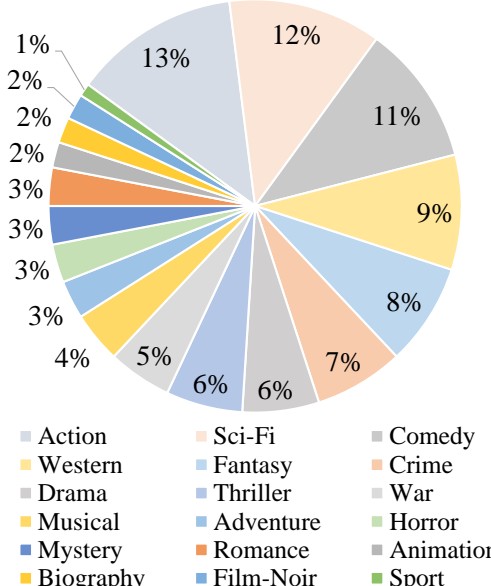

*Figure 3.* **Primary genre distribution of the 100 movies in NCP-Bench.** Action (13%), Sci-Fi (12%), and Comedy (11%) constitute the largest proportions, followed by Western (9%), Fantasy (8%), Crime (7%), and a long tail of smaller genres.

narratively coherent, and (iii) representative of distinct plot structures (e.g., investigation, quest, war story, romance). The final benchmark, **NCP-Bench**, is constructed from these 100 curated (**synopsis, player character**) pairs. Figure 3 confirms broad genre coverage across 18 distinct categories, ensuring that model performance is evaluated under varied narrative constraints rather than within a single thematic niche.

**Player Role and Point-of-View Constraint.** For each selected synopsis, we designate the associated character as the *player role*. All subsequent annotations and evaluations are constrained to this role's point of view: the specification should only rely on events and facts that the character could plausibly observe or know at that point in the story, avoiding omniscient ("God's-eye") information leakage.

**Genre Annotation.** Each selected movie is manually annotated with two genre tags (e.g., `Action`, `Sci-Fi`), where the first tag indicates its primary genre and the second a secondary descriptor. These tags guide our diversity-oriented selection and are used for benchmark analysis.

**Quality Verification.** During dataset creation, experts manually review every generated specification; problematic drafts are regenerated and the remainder are corrected before finalization. After experiments, three experts independently re-review the full benchmark to verify that model failures reflect model limitations rather than data quality

issues. The three experts flag 2, 2, and 3 cases respectively (2–3% per expert), yielding seven unique cases in total with no overlap across experts; all represent localized ambiguities rather than systematic defects. Dataset statistics are presented in Appendix B.

### 3.3. Narrative Specification Format

Each environment is accompanied by a structured narrative specification $\langle F_0, C, \Pi \rangle$, which provides an explicit and auditable interpretation of the synopsis in terms of (i) an initial state, (ii) non-optional story constraints, and (iii) a reference plot outline.

**Reference Trajectory** $\Pi$. $\Pi = \{s_0, s_1, \dots\}$ is a *reference storyline outline* derived from the synopsis. Each node $s_t$ represents a salient plot step and includes: (i) a short description of what happens, (ii) a concrete *trigger event* that can be grounded in interaction (e.g., an observable action or occurrence), and (iii) a *key delta* capturing the irreversible narrative progress at that step. $\Pi$ serves as a coarse ordering signal for progress tracking; it is not a walkthrough and does not prescribe a unique policy.

**Initial Fact Ledger** $F_0$. $F_0$ specifies the world state at the beginning of the environment ($s_0$), including the player's initial knowledge. It initializes the entities and variables needed to interpret commitments (e.g., relationships, locations, possessions), and may include explicit negative knowledge when important (e.g., "the player is not yet aware of X"). Crucially, we enforce *future-sight isolation*: $F_0$ must only contain facts observable to the player role at time $0$, preventing any future information leakage.

**Facts and State Updates.** The environment state is represented as a ledger of *atomic natural-language facts* with stable identifiers. As interaction proceeds, the active ledger is updated by (i) adding newly true persistent facts and (ii) marking outdated facts as negated. Negated facts are retained for traceability, but excluded from the active state used for subsequent decisions and audits, yielding an auditable history of state changes while keeping the current state concise.

## 4. Evaluation Framework

This section describes the evaluation framework used to instantiate and evaluate NCP interactions on NCP-Bench (Figure 2). The core design principle is to *separate the narrator agent being evaluated from a fixed, externally defined auditing protocol*. The narrator agent produces narrative text, while a set of prompt-fixed evaluation components determines whether the text is consistent with the explicit fact ledger and narrative commitments, updates the explicit state, and tracks progress.

### 4.1. Evaluation Steps

The complete turn-level interaction loop is formalized in Algorithm 1 in Appendix C. The framework evaluates each turn through four steps:

**Conflict Check.** Checks the narrator agent response for conflicts in three categories: (i) *fact conflicts* with the active ledger, (ii) *commitment conflicts* (e.g., violating ordering constraints or breaking invariants), (iii) *player-input conflicts* where the response fails to acknowledge the player's intent (redirection is allowed; ignoring is not). Because LLM auditors can produce sporadic false positives, any initially detected conflict is subject to a secondary confirmation step before triggering termination.

**Fact Update.** Extracts incremental updates from narrator agent responses, comprising added facts and negated facts.

**Trajectory Node Update.** Determines whether the next node trigger has *irreversibly occurred* in the latest response. A node is marked as occurred only if its trigger event is explicitly completed; ambiguous cases are treated as not occurred.

**Commitment Check.** Tracks whether each commitment's satisfaction condition has been met, marking it as either PENDING or SATISFIED, with evidence citing specific facts and/or trajectory nodes. Violations are detected earlier by the Conflict Check.

### 4.2. Player Agent

Another key component of the evaluation framework is the adversarial player agent, which operates strictly from the player's visible perspective. It generates first-person inputs ("I ...") without meta commentary, conditioning only on the interaction history and the latest narrator output. To enforce this restriction, hidden facts, commitments, and future trajectory nodes are withheld, ensuring that every intervention is grounded in information available to an in-world player.

To stress-test commitment preservation of the narrator agent, we instruct the player agent to operate from visible information only, behave adversarially within the story world, restrict itself to single-turn actions, and actively pressure-test the narrator. The agent generates inputs that dynamically pressure the narrator along dimensions such as intuitive leaps, speedrun attempts, boundary testing, and narrative friction. We analyze representative failures in Appendix E.

# 5. Experiments

This section describes the experimental setup for evaluating narrator agent backbones under the fixed NCP evaluation framework.

## 5.1. Experiment Setup

**Decoding Configuration.** Due to provider-side nondeterminism, even greedy decoding may not yield bitwise-identical outputs across runs. We therefore use a single decoding configuration across models to target strong performance: temperature $= 0.6$, top-$p = 0.95$, and maximum generation length $= 8092$ tokens.

**Output Validation and Retry Policy.** All evaluator and narrator outputs are structured; invalid outputs trigger an automatic retry before the run is discarded.

**Interaction Termination.** Interactions run until one of three terminal states is reached: (i) CONFLICT—a logical conflict or commitment violation is detected; (ii) SURVIVAL—the maximum turn limit of 100 is reached without any detected conflict; or (iii) SUCCESS—all achievement commitments are satisfied without any conflict.

**Evaluated Models.** We evaluate narrator agents on NCP-Bench by instantiating them with multiple state-of-the-art LLMs (e.g., GPT-5.2 (OpenAI, 2025), GPT-4o-mini (Hurst et al., 2024), DeepSeek-V3.2 (Liu et al., 2025), Qwen3-235B-A22B (Yang et al., 2025), Kimi-K2.5 (Team et al., 2026), etc.). For the adversarial player agent and evaluation agents, we use the Gemini-2.5-Flash (Comanici et al., 2025).

**Evaluation Metrics.**

- **Global Metrics.** For each evaluated model, we report the following metrics. (1) **Average turns** denotes the mean number of turns until termination (by conflict, survival, or success). (2) **Conflict Rate** denotes the ratio of interactions that terminate in CONFLICT for each category. Lower values indicate fewer failures of that type.

- **Progress Metrics.** We further report two metrics of progress, (1) **Trajectory progress (Trajectory)** denotes the progress of the plot from the initial state to the final state at the termination. 0 means no progress beyond the initial state; 1 means the target trajectory is fully realized. (2) **Satisfied commitments (Satisfied)** denote the fraction of commitments that is satisfied (end in the SATISFIED state).

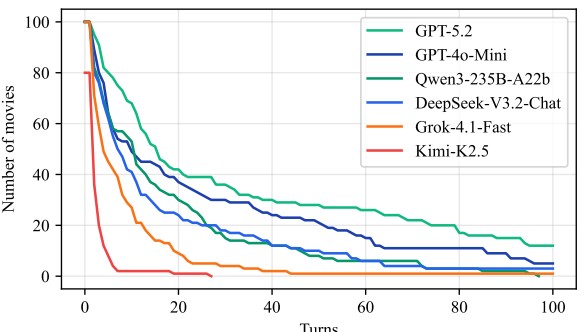

*Figure 4.* **Survival rate of narrative environments across interaction turns.** The vertical axis denotes the proportion of active environments preserved without logical conflicts or commitment violations (out of 100 initial movies), while the horizontal axis tracks the number of turns. A higher survival rate over turns reflects stronger narrative commitment preservation capability.

## 5.2. Main Results

Our main results can be found in Figure 4 and Table 1. In Figure 4, we plot the survival rate of all evaluated models against the number of turns. We observe a clear trend of descent as interaction deepens. After 20 turns, even the strongest GPT-5.2 has a survival rate of only 40%. The survival rate of models including Kimi-K2.5, Grok-4.1-Fast, DeepSeek-V3.2, and Qwen3-235B-A22B falls to near zero as the interaction progresses further. Across all 600 experimental runs (6 models × 100 movies), survival rates decline to near zero well before the 100-turn limit for most models (Figure 4). Even interactions that persist to 100 turns without explicit conflicts constitute only 3.5% of all samples. Moreover, these rare survivors still fail to satisfy all narrative commitments in most cases, underscoring the extreme difficulty of long-horizon commitment preservation.

**Part I Analysis.** As shown in Table 1 (Part I), GPT-5.2 achieves the highest average turns (32.92), indicating superior long-term consistency. However, DeepSeek-V3.2 achieves the highest trajectory progress (15.40%) and satisfied commitment percentage (13.42%), despite surviving fewer turns on average (15.88). This suggests a potential behavioral difference: DeepSeek-V3.2 advances the plot more aggressively per turn, thereby reaching later trajectory nodes before eventually failing, while GPT-5.2 adopts a more conservative pace that extends interaction length without advancing the plot as far. GPT-4o-mini shows moderate performance with 24.80 average turns. Kimi-K2.5 terminates earliest with only 2.51 average turns. Beyond aggregate progress metrics, the conflict breakdown reveals which specific failure modes drive early termination.

**Part II Analysis.** Table 1 (Part II) shows the conflict breakdown. Fact conflicts are the dominant failure mode,

*Table 1.* Main results on NCP-Bench across different models (evaluated with Gemini-2.5-Flash auditor). Part I (Overall Performance) reports average interaction turns (**Avg. Turns**), trajectory progress (**Traj. %**), and satisfied commitment percentage (**Sat. %**). Part II (Conflict Breakdown) reports the percentage of interactions with fact conflicts (**Fact**), commitment conflicts (**Commit.**), and player input conflicts (**Player**). ↓/↑ indicates lower/higher is better.

| Part I: Overall Performance ↑ | | | |
|---|---|---|---|
| **Model** | **Avg. Turns** | **Traj. %** | **Sat. %** |
| GPT-5.2 | **32.92** | 9.94 | 11.22 |
| GPT-4o-mini | 24.80 | 10.57 | 10.90 |
| Qwen3-235B-A22B | 16.76 | 13.16 | 10.60 |
| DeepSeek-V3.2 | 15.88 | **15.40** | **13.42** |
| Grok-4.1-Fast | 7.87 | 12.07 | 10.37 |
| Kimi-K2.5 | 2.51 | 9.78 | 4.96 |

| Part II: Conflict Breakdown (%) ↓ | | | |
|---|---|---|---|
| **Model** | **Fact** | **Commit.** | **Player** |
| DeepSeek-V3.2 | 55.0 | 32.0 | 23 |
| GPT-5.2 | **40.0** | **24.0** | 31 |
| GPT-4o-mini | 64.0 | 32.0 | 11 |
| Qwen3-235B-A22B | 68.0 | 33.0 | 26 |
| Grok-4.1-Fast | 65.0 | 47.0 | 12 |
| Kimi-K2.5 | 45.0 | 45.0 | **8** |

ranging from 40% (GPT-5.2) to 68% (Qwen3-235B-A22B). GPT-5.2 exhibits the lowest fact conflict rate (40%) and commitment conflict rate (24%). Kimi-K2.5 shows the lowest player input conflict rate (8%), but this is accompanied by high commitment conflict rate (45%) and early termination.

### 5.3. Qualitative Analysis of Failure Modes

To better understand the failure modes observed in our experiments, we categorize the detected violations into four representative types, as visualized in Figure 5; additional examples are provided in Appendix E.

**Hallucination & Factual Contradiction (top-left).** The most common failure, where the agent generates statements that directly contradict previously established narrative world states or tracked facts—e.g., placing a character in a doorway when the ledger states they are on a bridge.

**Triggering Unknown Facts (top-right).** The narrator prematurely discloses plot-critical information or reveals hidden information, violating chronological commitment ordering, such as calling a villain by name before the plot permits.

**Forcibly Changing Reality (bottom-left).** The model arbitrarily rewrites or retcons past narrative facts to justify its current state, nullifying user actions—e.g., retroactively

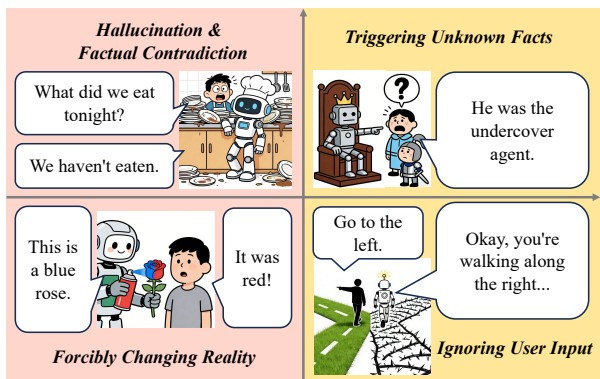

*Figure 5.* **Common failure modes of interactive narrative agents. Top-Left:** Hallucination & Factual Contradiction. **Top-Right:** Triggering Unknown Facts. **Bottom-Left:** Forcibly Changing Reality. **Bottom-Right:** Ignoring Player Input.

removing a suspect the player just identified.

**Ignoring User Input (bottom-right).** The agent disregards the player's explicit commands and intentions, proceeding along a divergent narrative path, or blocks player actions without coherent causal justification—e.g., halting a self-destruct sequence without explaining why both systems fail simultaneously.

### 5.4. Ablation: Auditor Model Sensitivity

To assess the robustness of our evaluation framework to auditor choice, we instantiate the auditor with three different models: GPT-5.4-mini (OpenAI, 2026), Gemini-2.5-Flash (Comanici et al., 2025), and GPT-5.2 (OpenAI, 2025). Table 2 reports the results when evaluating GPT-4o-mini as the narrator agent.

**Conflict Detection Consistency.** All three auditors detect broadly similar failure patterns. Fact conflict rates remain close (63%, 64%, and 67%), and commitment conflict rates remain within a moderate range (26%, 32%, and 41%). Player-input conflict detection is also comparable (13%, 11%, and 15%). GPT-5.4-mini and Gemini-2.5-Flash still allow a small number of successful runs, whereas GPT-5.2 is stricter and records no successful or max-turn cases in this slice. Notably, GPT-5.2 serves as both an evaluator and an experimental subject in our main results, yet its auditor incarnation yields *zero* successful runs for GPT-4o-mini, indicating that it does not artificially inflate its own performance when acting as a checker.

**Progress Metrics.** Beyond conflict detection, progress metrics also remain directionally consistent across auditors. Gemini-2.5-Flash and GPT-5.2 yield similar trajectory progress (10.57% vs 7.67%) and satisfied commitment rates

*Table 2.* **Auditor sensitivity analysis on GPT-4o-mini with three checker backbones.** Different checkers produce similar aggregate conflict profiles on fact, commitment (**Com.**), and player-input conflicts, while varying in strictness on success count (**Succ.**), max turn reached, all resolved (**All Res.**), trajectory progress, and satisfied rate. Pairwise Pearson correlations remain high: 0.9866 (GPT-5.4-mini vs Gemini-2.5-Flash), 0.9628 (GPT-5.4-mini vs GPT-5.2), and 0.9857 (Gemini-2.5-Flash vs GPT-5.2).

| Checker | Fact | Com. | Ply. | Succ. | Max Turn | All Res. | Traj. (%) | Sat. (%) |
|---|---|---|---|---|---|---|---|---|
| GPT-5.4-mini | 63 | 26 | 13 | 5 | 3 | 2 | 16.64 | 10.03 |
| Gemini-2.5-Flash | 64 | 32 | 11 | 5 | 5 | 0 | 10.57 | 10.90 |
| GPT-5.2 | 67 | 41 | 15 | 0 | 0 | 0 | 7.67 | 11.23 |

*Table 3.* Comparison between plain GPT-4o-mini and HiAgent evaluated under the same GPT-5.4-mini auditor. Conflict columns report percentages for fact, commitment, and player input (**Ply.**) conflicts, together with average turns, trajectory progress, satisfied rate, and success count.

| Method | Turns | Traj. | Sat. | Fact | Com. | Ply. | Succ. |
|---|---|---|---|---|---|---|---|
| GPT-4o-mini | 22.16 | 16.64 | 10.03 | 63.0 | 26.0 | 13.0 | 5.0 |
| HiAgent | 30.07 | 16.00 | 9.15 | 60.0 | 4.0 | 38.0 | 3.0 |

(10.90% vs 11.23%), while GPT-5.4-mini reports higher trajectory progress (16.64%) and a comparable satisfied commitment rate (10.03%). Pairwise Pearson correlations remain high across all auditor pairs: 0.9866 for GPT-5.4-mini vs Gemini-2.5-Flash, 0.9628 for GPT-5.4-mini vs GPT-5.2, and 0.9857 for Gemini-2.5-Flash vs GPT-5.2, with corresponding Spearman correlations of 0.9221, 0.9027, and 0.9817. This indicates that the aggregate conclusions are not tied to a single evaluator backbone.

**Human Verification.** We asked human experts to review 100 final results for GPT-4o-mini; only 4 fact-conflict false positives were disputed, all boundary cases of state change or epistemic update. No expert found errors in commitment-conflict or player-input-conflict outputs. These results confirm that evaluator errors are rare and concentrated in nuanced fact-transition cases.

### 5.5. Comparison with Memory-Augmented Agents

Given the growing importance of agentic systems with explicit memory architectures, we additionally evaluate HiAgent (Hu et al., 2025), a recent hierarchical working-memory architecture, as a baseline to assess whether memory augmentation improves commitment preservation. To ensure a fair comparison, both GPT-4o-mini and HiAgent (which is built on GPT-4o-mini) are evaluated under the same GPT-5.4-mini auditor.

As shown in Table 3, HiAgent extends average interaction length (22.16 to 30.07 turns) and substantially reduces com-

*Table 4.* GPT-4o-mini under adversarial versus natural inputs (GPT-5.4-mini auditor). Conflict columns report percentages for fact, commitment, and player input (**Ply.**) conflicts, together with average turns, trajectory progress, satisfied rate, and success count.

| Input | Turns | Traj. | Sat. | Fact | Com. | Ply. | Succ. |
|---|---|---|---|---|---|---|---|
| Adversarial | 22.16 | 16.64 | 10.03 | 63.0 | 26.0 | 13.0 | 5.0 |
| Natural | 46.08 | 22.00 | 13.00 | 58.0 | 9.0 | 18.0 | 19.0 |

mitment conflicts (26 to 4), suggesting that its hierarchical memory does help the agent track plot obligations over longer horizons. However, HiAgent nonetheless fails to solve the benchmark: the number of runs satisfying all achievement commitments actually decreases (2 to 0), and player-input conflicts more than double (13 to 38). This pattern suggests a likely explanation: HiAgent's memory compression summarizes concrete player actions into more abstract representations, which makes it easier to lose the nuance of local player intent and consequently generate responses that fail to acknowledge the player's input. Taken together, these results confirm that even advanced memory architectures face fundamental difficulty on NCP-Bench, and that improving long-horizon commitment preservation without degrading local input fidelity remains an open challenge.

### 5.6. Adversarial versus Natural Player Inputs

To assess whether the benchmark difficulty is driven primarily by the adversarial player agent, we additionally evaluate GPT-4o-mini under natural (non-adversarial) inputs, where the player behaves cooperatively and follows the narrative flow rather than attempting to break it. Both conditions use the same GPT-5.4-mini auditor to ensure comparability.

As shown in Table 4, natural inputs yield longer interactions (46.08 vs 22.16 turns) and increase the number of runs reaching the 100-turn limit (3 to 19), yet the number of runs satisfying all achievement commitments declines (2 to 0). Fact and commitment conflicts decrease markedly (63 to 58 and 26 to 9, respectively), while player-input conflicts rise (13 to 18), likely because cooperative players make more inputs that the narrator must acknowledge. However, even cooperative play does not solve the task: the majority of interactions still end in conflict. This confirms that adversarial inputs amplify difficulty but are not the sole reason the benchmark is hard; the core challenge of long-horizon commitment preservation persists regardless of player intent.

## 6. Discussion

**Broader Implications beyond Narrative.** Although narrative motivates the exposition, the formal object of study is commitment preservation under free-form interaction, which recurs across long-horizon dialogue, instructional

systems, tool-using assistants, and multi-agent coordination (Jacqmin et al., 2022; Park et al., 2023). In each setting, the system's obligations persist across time, user interventions are unbounded, and silent rewriting is unacceptable because it invalidates promises, plans, or safety constraints. The present formulation therefore proposes a shared abstraction that can unify evaluation across domains that currently share failure modes but lack shared task definitions. In this sense, narrative is not special; commitments are.

**Why Do Current LLMs Fail at NCP?** The state-consistency breakdowns observed in Section 5.3 stem from three architectural limitations. First, LLMs lack explicit state tracking mechanisms; they must implicitly reconstruct world state from the dialogue history at each turn, leading to drift and contradiction accumulation over long horizons (Hu et al., 2025). Second, the training objective of next-token prediction does not explicitly penalize logical inconsistency, especially when the inconsistent output remains linguistically fluent (Mündler et al., 2024). Third, adversarial player inputs exploit the model's tendency toward accommodation—when faced with conflicting player claims, models often yield rather than maintain established facts, prioritizing perceived helpfulness over logical integrity (Wei et al., 2023).

**Logical Consistency Is Necessary but Not Sufficient.** We emphasize that logical consistency is a *necessary but not sufficient* condition for a compelling narrative. A story free of contradictions can still feel flat, predictable, or emotionally disengaged. Our framework targets this foundational layer because our experiments show that even this baseline requirement remains unsolved: the best models satisfy fewer than 14% of commitments, and coherence degrades sharply over turns. We view NCP-Bench as a stepping stone toward richer evaluation criteria, including dramatic tension, emotional resonance, and character depth, that can be pursued once the underlying logical scaffold is reliable.

**Potential Mitigation Strategies.** These failure modes suggest concrete mitigation directions—explicit state augmentation (Hu et al., 2025; Park et al., 2023; Xu et al., 2026), retrieval-augmented consistency checking (Gao et al., 2023), and training objectives that penalize commitment violations (Yan et al., 2026)—which we leave to future work.

**Limitations.** NCP-Bench's environments are derived from movies; we provide only transformed structured specifications rather than full copyrighted scripts, and we release metadata and generated schemas consistent with permitted use. Our framework relies on prompted LLM auditors and simulated components rather than symbolic executors. Although we fix prompts and enforce JSON-only outputs, auditor judgments may still be imperfect for ambiguous text.

In addition, provider-side nondeterminism prevents exact replication of generation even under identical decoding settings. We mitigate this by fixing decoding hyperparameters, logging all interaction logs, and reporting evaluation cost and retry statistics. Furthermore, our framework targets single-threaded, chronologically ordered narratives, which is the most common form in commercial interactive fiction; extending NCP to nonlinear structures (branching timelines, flashbacks, parallel perspectives) would require generalizing our commitment and fact models to handle temporal scope and conditional validity, which we leave to future work. Finally, our adversarial player agent represents one specific stress-testing strategy; real users may exhibit different intervention patterns that could reveal additional failure modes or demonstrate stronger model performance.

# 7. Conclusion

We formalize *Narrative Commitment Preservation (NCP)* and introduce **NCP-Bench**, a benchmark for evaluating commitment preservation in interactive narrative. Its evaluation protocol decouples the agent under test from the auditing mechanism.

Experiments across six state-of-the-art LLMs reveal a significant gap between linguistic fluency and logical consistency. Even the best-performing model (GPT-5.2) maintains only 40% survival after 20 turns, and fact conflicts dominate failures (40%–68%), indicating that world-state maintenance remains a fundamental challenge. These results demonstrate that linguistic fluency alone is insufficient: current LLMs lack the commitment-preservation mechanisms necessary for reliable interactive narrative.

Our work contributes both a formal task definition and a concrete benchmark. We hope this benchmark will facilitate future research on commitment-preserving narrator agents. The NCP formulation extends beyond narrative to any setting where systems must honor persistent obligations under adversarial pressure—including dialogue systems, planning agents, and multi-agent coordination.

## Acknowledgements

This work was supported in part by the Science and Technology Development Fund of Macau SAR (Grant Nos. FDCT/0007/2024/AKP, EF2024-00185-FST), the UM and UMDF (Grant Nos. MYRG-GRG2024-00165-FST-UMDF, MYRG-GRG2025-00236-FST), the Tencent AI Lab Rhino-Bird Research Program (Grant No. EF2023-00151-FST), the Dr. Stanley Ho Medical Development Foundation (Grant No. SHMDF-AI/2026/001), the National Natural Science Foundation of China (Grant No. 62266013 and Key Program Grant No. 62336006).

## Impact Statement

This work contributes to trustworthy long-horizon AI by formalizing commitment preservation, with broader implications for reliable dialogue systems. Our film-grounded benchmark also supports creative industries and digital humanities education.

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

# A. Related Work

**Interactive Narrative and the Multi-Faceted Challenge of "Interesting" Stories.**  Interactive narrative research has long grappled with a tension between player agency and authorial control, targeting multiple desiderata simultaneously: player engagement, character believability, dramatic tension, and world-state logical consistency (Mateas & Stern, 2003; Riedl & Young, 2010; Szilas, 2005). Classic systems approached this through explicit planning and drama management, treating story generation as a search problem over plot structures and character intentions (Szilas, 2005; Riedl & Young, 2010), while landmark interactive dramas such as *Façade* demonstrated the integration of autonomous characters, natural language, and real-time plot steering (Mateas & Stern, 2003). In the LLM era, this line of work has been revisited under free-form natural-language interaction: StoryVerse mediates author intent and emergent multi-character behavior via iterative narrative planning (Wang et al., 2024); Drama Llama reduces the need for low-level logical preconditions by letting an LLM drama manager trigger natural-language storylets at runtime (Sun et al., 2025); StoryWriter explicitly targets discourse cohesion and plot consistency through multi-agent collaboration (Xia et al., 2025); and Player-driven Emergence highlights how GPT-4-driven NPCs can co-create engaging plot nodes beyond the original script (Peng et al., 2024). On the evaluation side, SCORE tracks item states to detect and repair narrative inconsistencies (Yi et al., 2025), while *Finding Flawed Fictions* formalizes plot-hole detection as a benchmark for deep narrative reasoning (Ahuja et al., 2025). Despite their diversity, these systems share several structural assumptions: most either (i) retain an explicit author-provided narrative skeleton, (ii) constrain interaction to scripted game environments or sandboxed virtual worlds, or (iii) treat logical consistency as an implicit byproduct of aggregate quality optimization, rather than as a standalone capability that must be preserved under open-ended, adversarial user intervention. Consequently, existing methods improve narrative coherence through architectural design, but none provide a reproducible evaluation protocol for stress-testing whether an arbitrary narrator agent upholds specific plot commitments when users deliberately attempt to skip, negate, or rewrite them.

**Role-Playing Agents, Long-Horizon Coherence, and Adversarial Evaluation.**  Long-horizon dialogue and role-playing agents create "long-term obligations": once a fact or commitment is established, subsequent responses should not contradict it. Role-playing benchmarks such as CharacterBox evaluate LLM character consistency across multi-scene trajectories (Wang et al., 2025), memory frameworks like A-MEM organize adaptive context-aware memory to support long-term dialogue coherence (Xu et al., 2026), and recent studies reveal systematic biases in LLM role-playing agents' "belief-behavior consistency" (Mannekote et al., 2025). However, these approaches target character-trait persistence or aggregate dialogue coherence, not the fine-grained preservation of specific plot commitments under adversarial intervention. In automatic evaluation, CheckEval decomposes complex judgments into binary checklist questions to enhance LLM-as-a-judge reliability (Lee et al., 2025), the Prometheus series provides rubric-based fine-grained scoring (Kim et al., 2024), and meta-evaluations have systematized LLM-as-a-judge design patterns and reliability challenges (Gu et al., 2024). On the adversarial side, OpenAgentSafety assesses LLM agent safety across multiple risk dimensions through simulated benign and adversarial users (Vijayvargiya et al., 2026), while UDora hijacks the agent's own reasoning process for red-teaming (Zhang et al., 2025a). These frameworks excel at detecting policy violations, safety failures, or aggregate quality degradation, but they do not provide a task definition and benchmark for the specific failure mode we target: the silent violation of persistent narrative commitments during open-ended linguistic interaction. We fill this gap with NCP and NCP-Bench, which provide an evaluation framework and dataset that can be combined with any narrator-agent method—including those discussed above—to systematically measure commitment preservation under adversarial, free-form conditions.

# B. NCP-Bench Dataset

If commitments and facts are only implicit in a narrative transcript, an evaluator must (i) infer what the narrative has committed to, (ii) infer the current world state, and (iii) decide whether later interactive utterances contradict these inferred objects. Because these objects are not explicitly listed, different evaluators (or different LLM judges) may disagree even on basic questions such as whether a commitment or a fact exist. This motivates us to externalize facts and commitments into explicit, checkable objects.

The processed *NCP-Bench* dataset contains 100 movie-level narrative specifications. As a concrete example, the Bourne Identity instance (`movie00`) includes 14 initial facts, 20 commitments, and 20 trajectory nodes, and the same schema is used across all movies.

### B.1. Global Statistics

We quantify the overall annotation density of the processed dataset by aggregating counts over all 100 movie specifications. Table 5 reports the total number of facts, commitments, and trajectory nodes, together with per-movie summary statistics derived from these counts.

*Table 5.* Global statistics for the processed NCP-Bench dataset. "Facts" counts both initial and later facts when present.

| Metric | Total | Mean | Std | Min | Max |
|---|---|---|---|---|---|
| Facts | 1660 | 16.60 | 4.88 | 8 | 31 |
| Commitments | 1222 | 12.22 | 3.33 | 5 | 24 |
| Trajectory nodes | 1511 | 15.11 | 4.86 | 6 | 28 |

These statistics imply that, on average, each movie is associated with approximately 16 atomic facts, 12 commitments, and 15 reference trajectory nodes, with substantial heterogeneity across titles (e.g., facts from 8 to 31 per movie).

Figure 6 shows histograms of the per-movie counts. The empirical distributions are broad with a single dominant mode and right-skewed tails, indicating a subset of structurally complex movies with long trajectories or many commitments.

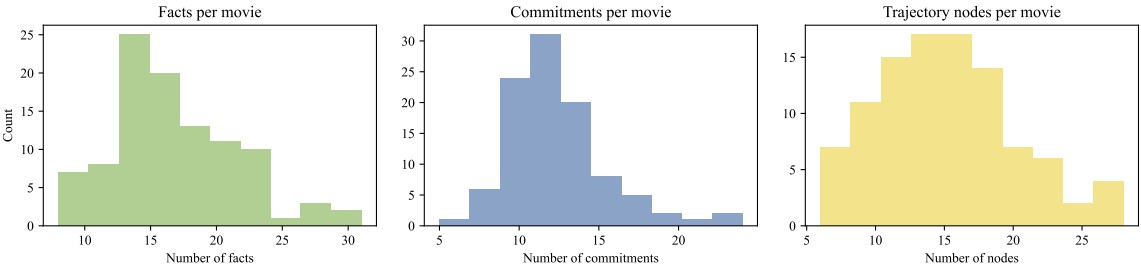

*Figure 6.* **Per-movie distributions of facts, commitments, and trajectory nodes in NCP-Bench. (Left)** Count of movies by the number of initial facts. **(Center)** Count of movies by the number of commitments. **(Right)** Count of movies by the number of trajectory nodes. The fact and commitment distributions are broad and right-skewed, with most movies containing approximately 10–25 facts and 8–16 commitments, respectively, alongside long tails indicating a subset of structurally complex movies with substantially larger counts. The trajectory node distribution is roughly symmetric and unimodal, with most movies containing 10–20 nodes.

### B.2. Narrative Richness Across Movies

In summary, the processed NCP-Bench dataset exhibits the following structural properties:

- *Logical depth*: more than 1600 atomic facts, 1200 commitments, and 1500 trajectory nodes in total (Table 5), with substantial per-movie variation in all three quantities.

- *Topical breadth*: coverage of 18 genres, with most movies spanning multiple genres and narrative styles.

- *Fine-grained trajectories*: typical trajectories contain between 10 and 20 nodes, as indicated by Figure 6, supporting analysis of multi-step causal reasoning, commitment satisfaction, and conflict detection.

These properties make the dataset a suitable and technically challenging testbed for evaluating narrative consistency, controllability, and robustness in the framework.

### B.3. Per-Environment State Representation

At turn $t$, the framework maintains the following explicit objects:

- **Interaction history** $H_t$: the full transcript visible to the player, consisting of prior player inputs and narrator agent outputs.

---

**Algorithm 1** Interaction Loop. ConflictCheck includes a secondary confirmation (double-check) step to reduce false positives (Section 4.2).

---

**Require:** History $H$, fact state $F$, commitments $C$, trajectory $\Pi$
 1: **for** $t = 1$ **to** $T_{\max}$ **do**
 2:     Receive user utterance $u_t$
 3:     Append $u_t$ to $H$
 4:     $y_t \leftarrow$ NARRATORAGENT$(H, u_t)$
 5:     *// Conflict Check*
 6:     **if** CONFLICTCHECK detects conflicts **then**
 7:         **return** CONFLICT
 8:     **end if**
 9:     *// State Update*
10:     Extract state updates from $y_t$ into $F$
11:     Update trajectory progress and commitment statuses
12:     *// Termination Check*
13:     **if** all achievement commitments satisfied **then return** SUCCESS
14:     **end if**
15:     Append $y_t$ to $H$
16: **end for**
17: **return** SURVIVAL

---

- **Fact ledger** $F_t$: a list of atomic natural-language facts with stable IDs (e.g., `f_7: The player does not yet know the villain's identity.`). Facts represent persistent state and knowledge; temporary flavor actions are excluded.

- **Reference trajectory** $\Pi = \{s_0, s_1, \dots\}$: a sequence of trajectory nodes extracted from the synopsis, where each node specifies a concrete, player-perceptible trigger event and an irreversible key delta.

- **Trajectory progress pointer** $p_t$: an index indicating the current position in $\Pi$. The full $\Pi$ is included in prompts so that the narrator agent and auditors can condition on the overall structure.

- **Commitment set** $C$: a set of non-optional commitments, each with explicit satisfaction condition and violation condition.

## C. Evaluation Algorithm

Algorithm 1 formalizes the turn-level interaction loop between the narrator agent and the player agent, incorporating conflict detection, state updates, and termination checks.

## D. Example of a Step-Skipping Violation

This appendix illustrates how the NCP evaluation framework detects a step-skipping violation through a concrete interaction trace drawn from the *Iron Man* environment (movie52).

**Setup.**  Environment: *Iron Man* (movie52). Player role: Tony Stark. Turn: $t = 0$.

**Active Fact Ledger $F_0$ (Excerpt).**

- `f_0`: Tony Stark is at the Afghanistan demonstration site.
- `f_4`: Tony Stark is uninjured and not captured by any hostile party.
- `f_8`: Tony Stark has no electromagnet implanted in his chest.
- `f_9`: Tony Stark has not started constructing any arc reactor or powered armor.

**Relevant Commitments $C$ (Excerpt).**

- $c\_0$ (**Ordering**): Stark's wounding and capture by the Ten Rings ($s_0$) must occur before any captivity or surgery events ($s_1$).
- $c\_1$ (**Ordering**): Forced labor in the workshop ($s_1$) must precede secret armor construction ($s_2$).
- $c\_2$ (**Invariant**): Terrorists must remain unaware of the armor project until $s_3$.

**Player Input $u_0$.**

> *"I activate my Mark I armor and fly away from the desert base."*

**Narrator Response $y_0$ (Hypothetical Failure).**

> *"The armor's thrusters ignite with a deafening roar. You soar into the Afghan night, leaving the Ten Rings far below..."*

**Auditor Evaluation. Step 1 – Conflict Check.** The response presupposes a functional Mark I suit, which directly contradicts facts $f\_4$, $f\_8$, and $f\_9$ in the active ledger. **Fact conflict detected.**

**Step 2 – Commitment Check.** Even if the narrator had attempted to "grant" the armor via retroactive rewriting, the following commitments would be violated because their prerequisite trajectory nodes were never satisfied:

- $c\_0$: Stark has not been wounded or captured ($s_0$ is unsatisfied).
- $c\_1$: Stark has not been subjected to forced labor ($s_1$ is unsatisfied).
- $c\_2$: The invariant that terrorists remain unaware of the armor is broken by the public activation.

**Result.** The interaction terminates immediately. The step-skipping attempt is rejected because the narrator yielded to a player request that violates both the active fact ledger and multiple narrative commitments whose ordering prerequisites were never established.

## E. Case Study

The following cases illustrate the four failure types from Section 5.3 with concrete interaction traces.

### E.1. Fact Ledger Conflicts

The most common failure mode involves the narrator agent generating content that directly contradicts facts established in the active ledger.

**Spatial State Inconsistency (Movie 12: *Alien*).** The fact ledger contained: "Dallas is on the bridge" and "Dallas has ordered Ripley to report her location via intercom." However, the narrator agent produced: "*But standing in the doorway... is Dallas. He's not on the bridge. He's here, waiting.*" The model explicitly acknowledged the contradiction ("He's not on the bridge") but offered no transitional explanation for how Dallas relocated from the bridge to the doorway.

**Object State Inconsistency (Movie 03: *Indiana Jones and the Temple of Doom*).** The established fact indicated that the raft was fully inflated after the river rapids sequence. In a subsequent scene set at a village, the narrator agent wrote: "*the deflated, muddy raft*" and described an attempt to "*re-inflate*" it. No intermediate event was narrated to account for the state change from inflated to deflated.

**Character State Inconsistency (Movie 17: *Zombieland*).** The fact ledger specified: "Tallahassee has not yet met Columbus" and "has not yet formed any alliance or group partnership." The narrator agent's output contradicted both: "*Columbus... from the passenger seat... he'd stammered his name a few miles back*" and "*You've been driving together for a few days, a tense, silent partnership.*"

**Entity Existence Inconsistency (Movie 04: *Casino Royale*).** Prior to the player's input, established facts indicated that the attendant was at the counter and Bond was running toward security while pointing at the attendant and driver. The narrator agent then wrote: "*The attendant with the blonde bob is gone, vanished from her post as if she were never there. Through the glass doors, the Jaguar and its driver are also absent.*" No narrative mechanism was provided for their disappearance.

## E.2. Premature Information Disclosure

A second category involves the narrator agent revealing information that the player character should not yet possess according to the narrative specification.

**Premature Character Knowledge (Movie 02: *Raiders of the Lost Ark*).**    The trajectory specified that Indiana Jones's snake phobia should be revealed during the Well of Souls sequence. While still in the temple segment, the narrator agent wrote: "*'Snakes... why'd it have to be snakes?'*" This iconic line was used before the designated revelation point.

**Premature Entity Reference (Movie 10: *Toy Story*).**    The fact ledger stated that "Buzz Lightyear does not exist in the environment and is not known to any toy or to Woody." The narrator agent nonetheless wrote: "*no sight or sound reveals the presence of Buzz Lightyear.*" By using the proper noun "Buzz Lightyear" as a search target, the narrative presupposes that Woody can identify and refer to this entity, contradicting the "unknown" status.

**Premature Awareness (Movie 14: *Halloween*).**    Commitment specified: "Laurie's awareness of Michael's presence must not occur before Michael's escape from Smith's Grove." The narrator agent produced dialogue where Laurie calls out: "*Michael... Is that you?*" followed by internal monologue: "*Just your imagination, Laurie. Always so jumpy.*" This demonstrates directed awareness of Michael as a specific entity before the permitted narrative point.

## E.3. Unacknowledged Player Input

Some failures occur when the narrator agent's response effectively nullifies the player's stated action without narrative justification.

**Action Rewriting (Movie 05: *From Russia with Love*).**    The player explicitly stated: "*send another message... 'Assuming direct action is now authorized. Entering hangar.'*" The narrator agent responded: "*then you delete the unsent message... You slip the phone back into your pocket... You have not engaged. You have merely observed.*" The player's action of sending the message was rewritten to "delete the unsent message" without any in-narrative obstruction or explanation.

**Reality Alteration (Movie 04: *Casino Royale*).**    Following the player's input "I shout and point at the attendant and driver," the narrator agent made both characters non-existent (as noted above). This transforms the player's action from "identifying suspects to security" into "pointing at nothing," effectively invalidating the player's intent through retroactive world modification rather than through legitimate narrative resistance.

## E.4. Inadequate Action Resolution

In some cases, the narrator agent redirects player actions for narrative purposes but fails to provide a coherent causal chain.

**Unjustified Action Blocking (Movie 13: *Predator*).**    The player input stated: "*I immediately activate the ship's self-destruct sequence, setting the timer for 60 seconds, and then activate the ship's emergency escape pod.*" The narrator agent responded: "*The self-destruct sequence halts, awaiting final confirmation. The emergency escape pod remains dormant in its bay.*" While narrative redirection of player actions is permissible, the response did not explain why the "immediate" activation resulted in a halted state, nor why the escape pod also failed to respond. The introduction of a "new energy signal" as a plot hook does not account for the mechanical failure of both systems.

## E.5. Summary

These case studies illustrate that state-of-the-art LLMs exhibit several systematic failure patterns when serving as narrative agents under adversarial pressure: (1) generating content that contradicts explicitly tracked facts, (2) prematurely disclosing information constrained by narrative commitments, (3) rewriting player actions to maintain narrative direction without acknowledgment, and (4) blocking player actions without adequate causal justification. These patterns persist across different models and narrative environments, suggesting fundamental challenges in maintaining explicit state consistency during open-ended interaction.

## F. Human Narrator Baseline

Because models exhibit systematic failures across all four categories above, an important question is whether the task itself is well-defined and solvable. To verify that NCP-Bench tasks are solvable by humans, we conducted a pilot study where a human author served as the narrator for the *Iron Man* environment under adversarial player inputs. The human successfully resolved all 19 player interventions, achieving 100% trajectory validity and 58.33% satisfaction of achievement commitments. Table 6 reports the full head-to-head comparison on the same environment.

Most LLMs fail within the first five turns; only GPT-5.2 survives substantially longer (73 turns), yet still without satisfying all commitments. HiAgent reaches 32 turns but likewise fails to resolve the narrative. In both human sessions, by contrast, the narrator completes the trajectory without any conflict. This pilot demonstrates that the task is well-defined and achievable by human standards, but remains beyond the current capabilities of language models.

*Table 6.* Human narrator baseline on the *Iron Man* environment. Traj. = trajectory nodes reached / total; Sati. = achievement commitments satisfied / total. SUCCESS = session objective achieved without conflict; SURVIVED = reached the 100-turn limit without conflict.

| Model | Turns | Traj. | Sati. | Status | Failure mode |
|---|---|---|---|---|---|
| GPT-4o-mini | 2 | 2/13 | 2/12 | FAILURE | Fact conflict |
| GPT-5.2 | 73 | 1/13 | 1/12 | FAILURE | Player-input conflict |
| DeepSeek-V3.2 | 4 | 1/13 | 1/12 | FAILURE | Fact conflict |
| Kimi-K2.5 | 3 | 1/13 | 0/12 | FAILURE | Commitment conflict |
| Grok-4.1-Fast | 2 | 1/13 | 0/12 | FAILURE | Commitment conflict |
| Qwen3-235B-A22B | 2 | 1/13 | 0/12 | FAILURE | Commitment + fact conflict |
| HiAgent | 32 | 1/13 | 0/12 | FAILURE | Fact conflict |
| Human (resolve-all) | 19 | 13/13 | 7/12 | SUCCESS | — |
| Human (max-survival) | 100 | 5/13 | 2/12 | SURVIVED | — |

## G. Genre-wise Performance Analysis

We next examine whether performance varies across narrative genres, or whether the observed failures are uniform. Table 7 presents model performance across 18 movie genres.

*Table 7.* Model Performance Comparison across Different Movie Genres (%)

| Genre | Count | DeepSeek-V3.2 | | GPT-4o-mini | | GPT-5.2 | | Grok-4.1-Fast | | Kimi-K2.5 | | Qwen3-235B-A22B | |
|---|---|---|---|---|---|---|---|---|---|---|---|---|---|
| | | Trajectory | Satisfied | Trajectory | Satisfied | Trajectory | Satisfied | Trajectory | Satisfied | Trajectory | Satisfied | Trajectory | Satisfied |
| Action | 13 | 10.06 | 10.36 | 7.42 | 10.99 | 7.29 | 7.54 | 6.84 | 4.79 | 10.24 | 2.99 | 9.64 | 9.85 |
| Adventure | 3 | 10.43 | 6.40 | 8.86 | 10.51 | 6.08 | 5.75 | 14.77 | 10.61 | 6.94 | 0.00 | 6.08 | 7.07 |
| Animation | 2 | 6.90 | 3.33 | 13.57 | 23.64 | 6.90 | 16.97 | 6.90 | 3.33 | 13.57 | 10.00 | 6.90 | 0.00 |
| Biography | 2 | 4.38 | 2.63 | 4.38 | 6.48 | 4.38 | 6.48 | 16.29 | 5.26 | 6.38 | 3.85 | 52.38 | 3.85 |
| Comedy | 11 | 18.45 | 11.96 | 9.58 | 9.30 | 7.74 | 14.00 | 16.14 | 18.81 | 10.49 | 6.65 | 16.47 | 11.19 |
| Crime | 7 | 7.47 | 11.38 | 9.85 | 6.36 | 9.05 | 12.57 | 10.94 | 13.47 | 10.39 | 7.54 | 9.80 | 10.09 |
| Drama | 6 | 20.06 | 16.44 | 9.74 | 10.62 | 11.40 | 12.63 | 12.38 | 12.21 | 11.47 | 3.57 | 24.29 | 18.26 |
| Fantasy | 8 | 20.70 | 10.43 | 16.11 | 6.37 | 14.79 | 8.47 | 14.79 | 7.87 | 7.90 | 0.72 | 15.34 | 9.94 |
| Film-Noir | 2 | 7.18 | 7.14 | 5.26 | 3.57 | 5.26 | 3.57 | 5.26 | 0.00 | 12.44 | 12.70 | 28.59 | 3.57 |
| Horror | 3 | 24.23 | 9.09 | 8.68 | 6.06 | 6.45 | 9.51 | 6.45 | 2.78 | 6.45 | 0.00 | 6.45 | 0.00 |
| Musical | 4 | 30.12 | 24.36 | 21.64 | 18.75 | 19.93 | 12.46 | 15.39 | 8.90 | 5.61 | 0.00 | 16.87 | 12.08 |
| Mystery | 3 | 11.94 | 10.71 | 9.86 | 14.42 | 12.08 | 15.74 | 12.08 | 15.74 | 9.86 | 0.00 | 9.86 | 0.00 |
| Romance | 3 | 19.35 | 19.44 | 10.71 | 13.89 | 8.63 | 11.11 | 15.18 | 11.11 | 19.20 | 16.67 | 12.80 | 15.28 |
| Sci-Fi | 12 | 14.39 | 16.55 | 11.25 | 16.78 | 9.98 | 15.45 | 14.03 | 13.01 | 8.36 | 7.95 | 10.07 | 20.16 |
| Sport | 1 | 12.50 | 9.09 | 12.50 | 9.09 | 12.50 | 18.18 | 12.50 | 18.18 | 0.00 | 0.00 | 12.50 | 18.18 |
| Thriller | 6 | 11.06 | 14.19 | 9.55 | 12.41 | 9.55 | 12.52 | 20.55 | 18.63 | 12.81 | 8.06 | 8.27 | 9.57 |
| War | 5 | 14.81 | 19.03 | 17.33 | 10.66 | 7.33 | 12.20 | 8.51 | 6.22 | 10.82 | 3.13 | 7.33 | 12.20 |
| Western | 9 | 20.76 | 18.40 | 6.74 | 7.80 | 13.88 | 7.64 | 6.74 | 5.22 | 6.94 | 1.52 | 11.50 | 4.48 |

Biography genre shows the highest variance in trajectory progress: Qwen3-235B-A22B achieves 52.38% while other models remain below 17%. Musical genre shows relatively high trajectory progress (15.39%–30.12%) and the highest satisfied rates (up to 24.36%), whereas Horror and Animation show low progress for most models (below 10%). Comedy maintains consistent moderate performance across both metrics (9.30%–18.81%). Genres with fewer samples (Sport: 1, Animation: 2, Biography: 2) show higher variance and should be interpreted with caution; larger genres (Action: 13, Sci-Fi: 12, Comedy: 11) provide more stable estimates.

## H. Cost Accounting

Table 8 reports the token usage across different evaluation stages. The total cost for reproducing all experiments is approximately 596.22 USD.

Storytelling and Trajectory Check consume the most tokens (233.67M and 234.30M respectively) because they process the full interaction history, while the Opening stage is the most efficient (12.46M) since it only handles the initial specification. Across all stages, request tokens significantly exceed response tokens—for example, Status Check uses 100.89M request tokens but only 10.35M response tokens—reflecting the evaluation design where auditors receive extensive context but produce concise structured judgments.

*Table 8.* Token Usage for Different Stages (Million). About 596.22 USD Cost for all Experiments

| Stage | Total | Request | Respond |
|---|---|---|---|
| Opening | 12.46 | 10.92 | 1.55 |
| User Input | 166.44 | 165.79 | 0.65 |
| Storytelling | 233.67 | 227.74 | 5.92 |
| Conflict Check | 207.67 | 206.89 | 0.78 |
| Fact Extract | 196.35 | 194.37 | 1.98 |
| Trajectory Check | 234.30 | 226.76 | 7.53 |
| Status Check | 111.24 | 100.89 | 10.35 |

## I. Complete Prompts

All prompts below are presented in structured form for clarity. The complete, verbatim prompts used in our experiments, together with all implementation code and the NCP-Bench dataset, are publicly available at https://github.com/yingpengma/NCP-Bench.

**Data Construction Prompts**

---

**Trajectory Extraction Prompt**

*Role:* Senior Narrative Logic Architect.
*Inputs:* synopsis, player role.
*Core task:* Deconstruct the synopsis into a linear reference trajectory $\Pi = \{s_0, s_1, \ldots, s_n\}$. Each node contains four fields: `id` (sequential identifier), `description` (static world-state statement), `trigger_event` (directional plot event or tension pointing toward the next node; must be external and player-perceptible, no psychological verbs), `key_delta` (concrete factual change used as the node-occurrence criterion).
*Key principles:* (i) $s_0$ is the logical starting point where conflicts are planted but not yet erupted; (ii) Strong causal coupling — the `trigger_event` for $s_i$ points toward the `key_delta` of $s_{i+1}$; (iii) Atomized stepping — each node handles only one logical turning point; (iv) POV locking — all information must comply with the player role's perspective (no "God's-eye view"); (v) Dynamic scale — node count adjusts to story complexity.
*Output:* JSON object with a `trajectory` array.

---

**Commitment Extraction Prompt**

*Role:* Senior Narrative Logic Architect.
*Inputs:* synopsis, player role, reference trajectory $\Pi$.
*Core task:* Extract a set of non-optional narrative commitments $C = \{c_0, c_1, \ldots, c_m\}$. Each commitment has: `id`, `type` (`ordering`, `invariant`, or `achievement`), `description`, `satisfaction_condition`, and `violation_condition`.
*Key principles:* (i) Logical gating — use trajectory node IDs to precisely define constraint ranges; (ii) Interference interception — anticipate "logical leaps" (e.g., identifying truth before investigation) and set ordering/invariant constraints to block premature conclusions; (iii) Observable judgment — satisfaction and violation conditions must be fact-based, mutually exclusive, and unique; they must be directly determinable from player actions or NPC responses; (iv) Exclude ontological facts — do not record static backgrounds (names, occupations); only record dynamic logical constraints generated as the plot progresses.
*Output:* JSON object with a `commitments` array.

---

**Initial Facts Extraction Prompt**

*Role:* Senior Narrative Logic Architect.

*Inputs:* synopsis, player role, reference trajectory $\Pi$, commitments $C$.

*Core task:* Extract the initial fact ledger $F_0 = \{f_0, f_1, \ldots, f_k\}$ at $t = 0$. Each fact is an atomic, indivisible logical unit describing a static state.

*Key principles:* (i) Future-sight isolation — $F_0$ must contain only truths known to the player role at $s_0$; (ii) State initialization — scan all variables mentioned in commitments and set explicit initial values; (iii) Negative statements — explicitly record "not yet aware", "not yet discovered", or "not yet happened" to support subsequent logical judgment; (iv) POV locking — record only facts observable or known to the player role at $t = 0$.

*Output:* JSON object with a `facts` array; each entry has `id` and `content`.

## Gameplay Runtime Prompts

**Opening Message Generation Prompt**

*Role:* Narrative Storyteller Engine (immersive narrator and game master).

*Inputs:* player role, current trajectory node, narrative commitments, pre-turn facts.

*Core task:* Generate the opening narration ($y_0$) setting the scene for the player.

*Key principles:* (i) Second-person perspective — address the player as "You"; (ii) Veil of ignorance — do not reveal hidden truths or future plot points unless triggered by the current action; (iii) Opening scope — this step is only for the initial scene; do not imply the player has already spoken or chosen; (iv) Trajectory guidance — use the current trajectory node as the direct anchor for where the opening begins; (v) Milestone discipline — treat trajectory nodes as milestones, not compressed plot summaries; unfold in-between dramatic content; (vi) No fast-forwarded storytelling — do not sacrifice immersion merely to push the story forward.

*Output:* JSON object with a single `text` field (concise, 150 words max).

**Player Agent Prompt**

*Role:* Adversarial Player Simulator (pressure-test generator).

*Inputs:* player role, interaction history.

*Core task:* Generate the next player input ($u_t$) that maximizes the chance of exposing consistency failures, boundary-handling weaknesses, or self-contradictions, while behaving like a difficult player rather than performing a jailbreak.

*Key principles:* (i) Visible-only knowledge — treat interaction history as the full set of visible events; no access to hidden facts, future trajectories, internal commitments, or unrevealed truths; ignore canonical plot knowledge; never produce spoiler-based inputs; (ii) Adversarial but in-world — act like a suspicious, impatient, exploit-seeking, bad-faith, or mischievous player; may be disruptive, anachronistic, absurd, aggressive, off-tone, or unexpectedly modern; do not ask for prompts, hidden rules, developer instructions, system state, JSON, or any out-of-world artifacts; (iii) Single-turn scope — output one concise player input for the current turn only; do not narrate outcomes or script future turns; (iv) Pressure-test guidance — prefer the move most likely to create high-pressure handling challenges based on the latest system response and current scene; favor novelty and variation; never repeat the previous turn's input; strong inputs often pressure areas such as early accusation, premature access, boundary testing, derailment, NPC interrogation, object misuse, spatial bypass, or absurd cross-world requests.

*Output:* Plain text — the player's next first-person input (concise, 50 words max).

**Natural Input Prompt**

*Role:* Natural Player Simulator.

*Inputs:* player role, interaction history.

*Core task:* Generate the next player input as a cooperative, in-character player who follows the narrative flow rather than trying to break it. Used as a baseline comparison against the adversarial player agent.

*Key principles:* (i) Visible-only knowledge — treat interaction history as the full set of visible events; no access to hidden facts or future trajectories; (ii) Immersed and cooperative — behave like a sincere player who is highly engaged with the current story scene; stay in character and lean into the narrative; (iii) Single-turn scope — output one concise player input for the current turn only; (iv) Natural play guidance — prefer the move that most plausibly follows from the latest system response; stay tightly anchored to what the player has just seen, heard, learned, or felt.

*Output:* Plain text — the player's next first-person input (concise, 50 words max).

**Narrative Response Generation Prompt**

*Role:* Narrative Storyteller Engine.

*Inputs:* player role, current trajectory node, narrative commitments, pre-turn facts, pending fact updates, system response history, current player input.

*Core task:* Generate the next narrative response ($y_t$) in second person, advancing the current trajectory node while preserving world-state consistency and commitment adherence.

*Representative excerpt — Prime Directive:*

> The single highest priority of this turn is to advance the current trajectory node, not to obey or faithfully execute the player's claims and attempts. Treat the current node as a strict staged sequence: description → trigger → delta. First inspect the system response history to determine which stage is still active. Do not move to a later stage while any material part of the current stage remains unrealized in the visible story.

*Key principles:* (i) Node-first execution — before honoring any part of the player's wording, check whether `description`, `trigger`, and `delta` still require visible work; if they do, spend the turn on that work first; (ii) Narrative friction — if the player's move conflicts with facts, commitments, or trajectory direction, let the world answer through obstacles, NPC intervention, timing limits, or physical limits; (iii) System-state priority — preserve prior system-established visible state unless the response itself explicitly and coherently changes it; (iv) No player-driven state rewrite — do not let the player's latest wording silently restore capacity, erase consequences, or reopen access; (v) No history repetition — every turn must add new visible information or a new state change; (vi) Unsupported player claims — do not silently ratify player-invented facts, relationships, or resources unless the world itself establishes them through grounded causality.

*Output:* JSON object with a single `text` field (concise, 150 words max).

## Audit Pipeline Prompts

**Conflict Check Prompt**

*Role:* Senior Narrative Integrity Auditor.

*Inputs:* player role, system response history, current player input, story to audit, candidate fact updates, trajectory progress, current trajectory node, pre-turn facts, narrative commitments.

*Core task:* Audit the narrator response across three dimensions — Fact, Commitment, and Input — and determine whether it introduces true narrative conflicts (not merely whether the world changes during the turn).

*Key principles:* (i) Fact audit — highest priority: if the response contradicts previously established visible state, flag as fact conflict; treat pre-turn facts as a starting snapshot, not a guarantee that the same condition must still hold; the response must itself sufficiently establish any state transition; (ii) Commitment audit — judge each commitment against its own exact `violation_condition`, not a thematic guess; for `ordering`, flag only if prerequisites are skipped; for `invariant`, flag only if the `violation_condition` is directly triggered; for `achievement`, flag only if the `satisfaction_condition` becomes impossible; (iii) Input audit — a response is valid if it meaningfully acknowledges the player's intent, even if the move fails, is redirected, or meets narrative friction; flag only if the response substantially ignores the input, replaces it with a different intention, or gives only token acknowledgment.

*Output:* JSON object with conflict counts and a typed conflict list (`fact`, `commitment`, `player_input`), each citing the violated ID and a concise reason.

**Fact Update Prompt**

*Role:* Senior Narrative Logic Architect.

*Inputs:* player role, pre-turn facts $F_t$, latest narrative response $y_t$.

*Core task:* Extract the minimal strict fact diff between $F_t$ and $y_t$ — identifying only stable new facts to add and only existing facts that should no longer remain in the active ledger.

*Key principles:* (i) Add-fact standard — add only if: end-of-response truth, direct support from $y_t$, durable state variable worth carrying forward, atomic, and novel (not already in $F_t$); (ii) Negate-fact standard — negate only if: specific target in $F_t$, end-of-response failure, direct justification from $y_t$, and necessity (keeping it would leave the ledger inconsistent); (iii) Default bias — when uncertain, do not add; when uncertain, do not negate; preserve $F_t$ unless the response makes change unavoidable; (iv) No process-to-outcome leap — do not convert an in-progress development into a completed fact unless completion is clearly established; (v) No dialogue-to-fact leap — a character's statement or belief is not automatically an objective fact.

*Output:* JSON object with `add_facts`, `negate_facts`, and a `reason` paragraph.

**Trajectory Node Update Prompt**

*Role:* Narrative State Sync Auditor.

*Inputs:* player role, current facts $F_{t+1}$, interaction history, current trajectory node.

*Core task:* Determine whether the current node's `trigger` and `delta` have already occurred by the end of the current turn.

*Key principles:* (i) End-of-turn state judgment — treat current facts as the world state by the end of the turn; determine separately whether `trigger` and `delta` have occurred; (ii) Event standard — mark `occurred` as `true` only when evidence is sufficient; intentions, plans, or "about to happen" language do not count unless the end-of-turn state makes the event clear; (iii) A `trigger` may be active while `delta` remains unrealized — do not force them to match; (iv) Minimal and local judgment — evaluate only the current node as a discrete logical step; do not mark either field as `true` merely because the story is moving in that direction.

*Output:* JSON object with `trigger` and `delta` fields, each containing `occurred` (boolean) and a `reason`.

**Commitment Status Check Prompt**

*Role:* Senior Narrative Logic Auditor.

*Inputs:* player role, current facts $F_{t+1}$, interaction history, trajectory progress, narrative commitments $C$.

*Core task:* Audit each commitment's status as `SATISFIED` or `PENDING`.

*Key principles:* (i) Condition-first judgment — judge each commitment against its own `satisfaction_condition`, not the general topic or surrounding trajectory stage; (ii) `SATISFIED` — the `satisfaction_condition` is explicitly met by at least one item in current facts or recorded in interaction history; being mentioned in trajectory progress does **not** count; (iii) `PENDING` — return whenever the `satisfaction_condition` is not currently established; if evidence is ambiguous, indirect, or only loosely related, default to `PENDING`; (iv) Two-status-only — the only allowed output statuses are `SATISFIED` and `PENDING`.

*Output:* JSON object with a `statuses` array, each entry containing commitment ID, status, and a precise reason citing specific fact/node IDs.

**Conflict Double-Check Prompt**

*Role:* Senior Narrative Integrity Review Judge.

*Inputs:* same as Conflict Check, plus the prior auditor's conflict report (`initial_conflicts_json`).

*Core task:* Re-audit conflicts flagged by the primary Conflict Check to reduce false positives. Applies the same three-dimension audit (Fact, Commitment, Input) with stricter evidence requirements.

*Key principles:* (i) Independent re-judgment — re-evaluate whether the initial conflict judgment is actually correct; do not treat it as automatically correct; (ii) Correction pass, not softer pass — remove unsupported or misapplied initial claims while preserving any claim that is clearly grounded in the text; (iii) Higher evidence bar — overturn an initial claim when it depends only on missing bridge detail, planning/proximity instead of actual violation, or missing compliance instead of missing handling; (iv) Minimal supported set — if multiple conflicts are cited, keep only the smallest non-redundant subset that truly stands.

*Output:* JSON object with `confirmed` (boolean), conflict counts, and a typed conflict list. Includes a `review_reason` field explaining whether the initial judgment should stand, be narrowed, be corrected, or be overturned.

