# OpenReview forum: "Can LLM Agents Stick to the Script?  Modeling Commitment in Interactive Narratives"
_ICML.cc/2026/Conference — ICML 2026 regular_

### Official Review · Reviewer_vkjN · 2026-03-07

**Soundness:** 2
**Presentation:** 2
**Significance:** 3
**Originality:** 4
**Overall Recommendation:** 5
**Confidence:** 4

**Summary:**

The paper addresses the core challenge in the field of Interactive Narrative: maintaining logical consistency and narrative integrity under open, free-form user interventions. To this end, the authors formulate the Narrative Commitment Preservation task and introduce the NCP-Bench dataset. This framework models interactive narrative as a long-horizon constraint satisfaction problem, enabling automated and reproducible evaluation of narrative agents' logical consistency through explicit maintenance of the Fact Ledger and Narrative Commitments, coupled with fixed LLM-based auditing. Experiments covering six state-of-the-art LLMs, including GPT-5.2 and DeepSeek-V3.2, reveal that even the strongest model achieves only a 40% survival rate after 20 turns of adversarial interaction, with fact conflicts constituting 40% to 68% of failure modes, highlighting the significant gap between current LLMs' logical commitment preservation and linguistic fluency.

**Compliance With Llm Reviewing Policy:**

Affirmed.

**Final Justification:**

Based on the authors' comprehensive rebuttal, I upgrade my rating from 4 to 5. The authors have directly addressed all my concerns with concrete evidence: (1) They added cross-backbone validation using GPT-5.4-mini, showing high consistency across three evaluators (Pearson correlations 0.9628-0.9866, Spearman 0.9027-0.9817) and conducted human expert review of 100 final results with only 4 disagreement cases, all being fact conflict false positives; (2) They clarified that benchmark specifications underwent manual expert checking, with problematic drafts re-tried and retained specifications manually corrected, plus an additional independent review by 3 experts flagging only 7 movies (2.3% per-expert rate) as localized ambiguity rather than widespread inconsistency; (3) They acknowledged the "update commitments" terminology was imprecise and will clarify that the commitment set remains static while statuses transition; (4) They will add explicit figure references in the main text; (5) They explicitly frame nonlinear narrative as a scope limitation and future direction. These revisions demonstrate rigorous evaluation standards, terminological precision, and responsible scope acknowledgment, warranting a stronger accept recommendation.

**Key Questions For Authors:**

1. Figures 1 and 2 are not referenced or introduced anywhere in the main text of the paper.
2. Commitments, trajectories, and facts are extracted from movie synopses via LLM prompts. How is their quality ensured? Has any human validation been conducted?
3. In Algorithm 1 step 10, the text states "update commitments," but Section 3.1 explicitly states that "The commitment list is static among the whole interaction." This appears contradictory. Would it be more accurate to state "assigns each commitment a status" or "updates commitment status"?
4. How are flashback and nonlinear narrative structures handled?
5. In the ablation study, both Gemini-2.5-Flash and GPT-5.2 are used as checkers. Is there any comparison against human validation? Furthermore, since GPT-5.2 serves as both a checker and an experimental subject, could bias lead to artificially inflated performance for GPT-5.2?

**Limitations:**

Yes

**Strengths And Weaknesses:**

Strengths：In interactive narrative, logical collapse caused by hallucinations represents a critical bottleneck for practical deployment. This work transforms the vague notion of coherence into a checkable commitment preservation problem, providing a novel evaluation standard for the field. The authors construct the NCP-Bench dataset comprising 100 movies spanning 18 genres, design a separated evaluation framework, and implement automatic conflict detection through four specialized auditors.
Weaknesses：The reliability of the auditors themselves lacks human validation. Although the authors conduct ablation experiments with two LLM auditors, both are LLMs and may share similar biases. For ambiguous narrative boundary cases, such as the distinction between implication and explicit statement, different auditors may disagree, yet the paper does not report the auditing accuracy on a human-annotated subset. Figures 1 and 2 lack explicit references and detailed introductions in the main text, requiring readers to infer their connections to the text.

---

> ### Author Rebuttal · Authors · 2026-03-31
>
> **We thank you for your positive evaluation and for commenting on our work targeting a critical bottleneck and NCP-Bench as a novel evaluation standard for the field.**
>
> ## Reliability of the evaluator and lack of human validation & GPT-5.2 as both evaluator and experimental subject
>
> We agree and added two direct checks: an additional evaluator backbone (GPT-5.4-mini) and human validation of final results.
>
> With GPT-4o-mini as the storyteller, GPT-5.4-mini, Gemini-2.5-Flash, and GPT-5.2 produce highly consistent aggregate outcomes. Based on the three-evaluator summary table, the pairwise Pearson correlations (GPT-5.4-mini / Gemini-2.5-Flash, GPT-5.4-mini / GPT-5.2, Gemini-2.5-Flash / GPT-5.2) are 0.9866, 0.9628, and 0.9857, and the corresponding Spearman correlations are 0.9221, 0.9027, and 0.9817.
>
>
> |Evaluator|Fact conflict|Commit. conflict|User conflict|Success count|Max-turn survival|All-resolved|Trajectory process %|Satisfied process %|
> |:-------:|:-----------:|:--------------:|:-----------:|:-----------:|:---------------:|:----------:|:------------------:|:-----------------:|
> |GPT-5.4-mini|63|26|13|5|3|2|16.64|10.03|
> |Gemini-2.5-Flash|64|32|11|5|5|0|10.57|10.90|
> |GPT-5.2|67|41|15|0|0|0|7.67|11.23|
>
> We also asked human experts to review 100 final results on GPT-4o-mini.
> The union of disagreements contains only 4 cases, all of which are fact conflict false positives. For example, misjudging reasonable space transitions, plot changes, or cognitive updates, or mistaking a character's cognition for objective facts. No expert found errors in final results with commitment conflict or user input conflict.
>
> Taken together, these results indicate that the evaluator is stable across evaluator backbones and that the observed residual errors are rare and concentrated in a narrow class of nuanced fact-transition cases.
>
> ## Quality control for the benchmark specifications
>
> The benchmark specifications were not used in a purely automatic form. In the original construction pipeline, the extracted specifications were manually checked by experts, problematic drafts were re-tried, and the retained specifications were manually corrected before finalization.
>
> During the rebuttal period, we organized an additional independent review by 3 human experts. Their union flagged 7 movies as potentially problematic, but no item was independently flagged by two or more experts. This corresponds to an average per-expert flagging rate of 2.3%. These cases are localized ambiguity or alignment issues, rather than widespread benchmark inconsistency.
>
> Therefore, while these cases indicate that some specifications can still be improved, they do not suggest that the main benchmark defects.
>
> ## “Update commitments” versus static commitment lists
>
>
> Thank you for pointing this out.
> The commitment list remains fixed during interaction; what changes is each commitment's status. That is, the commitment set is static, while their statuses transition among pending, satisfied, and violated.
>
> We will refine our text to make the logic clearer.
>
> ## Figure references in the main text
>
> Thank you for your careful reading. We will add explicit descriptive text and reference in the main text in our next version.
>
>
> ## Flashbacks and nonlinear narrative structure
>
> We are deeply grateful for this perceptive observation. The reviewer’s insight highlights an important dimension of narrative complexity that deserves careful consideration.
> In the current version of NCP-Bench, we have centered the benchmark on auditable world-state evolution and mandatory narrative commitments rather than a literal encoding of editing order.
>
> We fully agree that explicitly modeling nonlinear presentation order would be a valuable extension. We will explicitly frame this as a scope limitation in the revised paper, and discuss it as a promising direction for future iterations of the benchmark.
>
> **In summary, we have added cross-backbone validation and human verification to address evaluator reliability concerns, clarified the static-commitment terminology to prevent misreading, and acknowledged nonlinear narrative as a future direction beyond current scope, while confirming that benchmark specifications underwent expert review. These revisions directly address your concerns regarding evaluation rigor, terminological precision, and benchmark coverage.**
>
> **We hope these results resolve your concerns, and thank you again for your support of our work.**

---

### Official Review · Reviewer_EV6E · 2026-03-12

**Soundness:** 3
**Presentation:** 3
**Significance:** 3
**Originality:** 3
**Overall Recommendation:** 4
**Confidence:** 3

**Summary:**

This paper examines the inability of current Large Language Models (LLMs) to maintain logical consistency and adhere to established narrative constraints under unconstrained user interventions in AI-driven interactive storytelling. The authors formalize this challenge as "Narrative Commitment Preservation" (NCP). The problem is defined as casting interactive narrative as a state-tracking and constraint satisfaction problem. A benchmark named NCP-Bench is introduced, which contains 100 diverse narrative environments derived from movie synopses, annotated with structured specifications (reference trajectories, commitments, and initial facts).

An extensive empirical evaluation of six state-of-the-art LLMs (including GPT-5.2, GPT-4o-mini, and DeepSeek-V3) interacting with an adversarial player simulator for up to 100 turns. The study systematically analyzes survival rates, progress metrics, and specific conflict types (e.g., fact contradictions, user input ignorances), demonstrating that all evaluated models fail to consistently preserve logical commitments over extended interactions, with no model successfully completing a full narrative.

**Compliance With Llm Reviewing Policy:**

Affirmed.

**Final Justification:**

The rebuttal has addressed my concerns. I would like to keep my score on the positive side.

**Key Questions For Authors:**

1. To what extent were the automatically generated narrative specifications (reference trajectories, commitments, and initial facts) manually verified or corrected by human experts? How to ensure that the "failures" detected by the auditor are not due to contradictions or ambiguities within the benchmark's own ground-truth specifications?
2. Did the authors conduct any human pilot studies to establish a success rate for humans against the same adversarial player simulator? Without a human baseline, it is difficult to determine if a 100-turn survival rate against an  adversary is a realistic expectation for any agent, or if the task is structurally designed to guarantee failure.

**Limitations:**

yes

**Strengths And Weaknesses:**

**Strengths**
1. This paper addresses a central concept in the deployment of AI agents: maintaining robust, logically consistent world models over long, unconstrained interactions.
2. The formulation of NCP as a long-horizon constraint satisfaction problem provides a rigorous and objective means for evaluating the functional reliability of interactive narratives.
3. The evaluation methodology is well-designed. It employs an adversarial player simulator to actively stress-test the storytellers, thereby avoiding the pitfalls of passive or overly cooperative evaluations that can mask model weaknesses.

**Weaknesses**
1. The empirical study focuses exclusively on general-purpose, off-the-shelf LLMs. It lacks comparison against models equipped with explicit state-tracking mechanisms, RAG methods, or architectures specifically designed for constraint satisfaction/narrative generation. Without such baselines, it is difficult to determine if the failures stem from the limitations of autoregressive models or simply the lack of appropriate augmentations for this specific task.
2. The NCP-Bench environments are derived merely from linear movie plots, which inherently contradicts with the interactive dynamics and branching possibilities. Attempting to force a highly constrained linear narrative onto an open-ended interactive environment with an adversarial player might structurally cause failure.

---

> ### Author Rebuttal · Authors · 2026-03-31
>
> **We sincerely appreciate the reviewer's validation of our NCP task formulation and for recognizing us for solving a central concept in AI agent deployment.**
>
> ## Missing comparisons with stronger baselines
>
> We added HiAgent (ACL 2025) as a stronger baseline. Both GPT-4o-mini and HiAgent are evaluated with the same GPT-5.4-mini evaluator.
>
> |Method|Count|Avg turns|Avg traj.|Avg sati.|Success count|Max turn survival|All resolved|Fact conflict|Commit. conflict|User conflict|
> |:---:|:---:|:---:|:---:|:---:|:---:|:---:|:---:|:---:|:---:|:---:|
> |Baseline|100|22.16|0.17|0.10|5|3|2|63|26|13|
> |HiAgent|100|30.07|0.16|0.09|3|3|0|60|4|38|
>
> HiAgent increases average turns and reduces commitment conflicts (26→4), but fact conflicts barely decrease (63→60), success drops (5→3), and user-input conflicts rise sharply (13→38). Under memory compression, HiAgent tends to abstract concrete player actions, making it easier to drift from user intent. This shows that stronger scaffolding still does not eliminate core failure modes.
>
> ## Whether linear movie plots structurally conflict with interactive narrative
>
> NCP-Bench does not require literally replaying a movie scene by scene. It constrains only narrative invariants—active facts, explicit commitments, and user inputs—while leaving local dialogue and intermediate actions open. All decisive factors are explicitly available in context; there is no hidden component. The benchmark tests commitment preservation under interaction, not obedience to a fixed script.
>
> We also compared adversarial vs. natural user inputs. Under natural input, performance improves substantially (avg turns 22→46, success 5→19, commitment conflicts 26→9), but the task remains unsolved (100-turn survival: 0). This suggests adversarial inputs amplify difficulty but are not the sole reason the benchmark is hard.
>
> |Method|Count|Avg turns|Avg traj.|Avg sati.|Success count|Max turn survival|All resolved|Fact conflict|Commit. conflict|User conflict|
> |:---:|:---:|:---:|:---:|:---:|:---:|:---:|:---:|:---:|:---:|:---:|
> |Adversarial|100|22.16|0.17|0.10|5|3|2|63|26|13|
> |Natural|100|46.08|0.22|0.13|19|19|0|58|9|18|
>
> ## Specification quality and human re-review
>
> Specifications were manually checked by experts during dataset creation. During rebuttal, 3 independent human experts re-reviewed all 100 movies for contradictions, over-constraints, and misalignment. The union of flagged cases contains 7 movies; no movie was flagged by more than one expert (average per-expert flag rate: 2.3%). Flagged issues are localized wording/scope concerns, not systematic benchmark defects.
>
> ## Human baseline
>
> We added a human pilot on Iron Man with two sessions. In one (resolve-all), the human reached 100% trajectory completion and 58.33% commitment satisfaction in 19 turns conflict-free. In another (max-survival), the human survived all 100 turns without any conflict. `✓`/`✗` indicate no/presence of conflict; Status `✓` means successful.
>
> |Model|Turns|Traj.|Sati.|Fact|Commit.|User|Status|
> |:---:|:---:|:---:|:---:|:---:|:---:|:---:|:---:|
> |GPT-4o-mini|2|2/13|2/12|✗|✓|✓|✗|
> |GPT-5.2|73|1/13|1/12|✓|✓|✗|✗|
> |DeepSeek-v3|4|1/13|1/12|✗|✓|✓|✗|
> |Kimi-k2.5|3|1/13|0/12|✗|✗|✓|✗|
> |Grok-4.1-fast|2|1/13|0/12|✗|✗|✓|✗|
> |Qwen3-235b-a22b|2|1/13|0/12|✗|✗|✗|✗|
> |HiAgent|32|1/13|0/12|✗|✓|✓|✗|
> |Human (all resolved)|19|13/13|7/12|✓|✓|✓|✓|
> |Human (max survival)|100|5/13|2/12|✓|✓|✓|✓|
>
> All LLM-based methods either failed early or survived longer only by making little progress. The task is not structurally impossible—there is a substantial gap between current LLMs and human performance.
>
> **In summary, we added HiAgent showing improved scaffolding still fails on core modes; clarified the benchmark tests commitment preservation, not script obedience; confirmed specification issues are localized (2.3%); and demonstrated task feasibility via human pilot.**
>
> **We hope these results resolve your concerns, and thank you again for your support of our work.**

---

> > ### Author Rebuttal · Reviewer_EV6E · 2026-04-03
> >
> > The authors’ response has resolved my concerns. I would like to keep my initial ratings for now.

---

> > > ### Author Response · Authors · 2026-04-03
> > >
> > > Thank you for your response and for carefully considering our rebuttal. We are pleased that our clarifications and additional results resolved your concerns. We sincerely appreciate your constructive feedback and your support for our submission.

---

### Official Review · Reviewer_MDPm · 2026-03-12

**Soundness:** 3
**Presentation:** 3
**Significance:** 2
**Originality:** 2
**Overall Recommendation:** 5
**Confidence:** 4

**Summary:**

The context of this work is in the following issue: LLMs tend to not generate consistent narratives. They can hallucinate facts and even contradict earlier parts of a story/quest/narrative they generated.

The focus of this work is provide a more robust framework in which one can evaluate the quality of an LLMs narrative. A scheme is proposed to track progress of a story through a "trajectory". The expectation is that the "storyteller" (e.g. LLM generating the story) will produce a narrative response to a user's response (or "utterance", as it is called in the work). A trace of such a process is checked in each step for integrity (i.e. conflicts) and to see if all global constraints/commitments are adhered to. More formally, a history of user response / storyteller response pairs is tracked along with a given set of facts at each time (parts of the story), commitments, and progress in the story (i.e. "trajectory").

The next major contribution of this work is experiments performed using a number of LLMs in this framework. Figure 4 in particular shows the success of various LLMs in maintaining a coherent/consistent narrative over a sequence of turns: notably it drops off sharply. Table 1 describes the frequency with which an LLM reached one of the "satisfied" conditions for the story (i.e. successfully concluded the story without any conflicts). Unsurprisingly, these were all very low as well with the best performance being under 14%. A potentially-helpful summary of the various reasons for failure is provided. While not the main point of this paper, it could help other work address the problem of the poor performance of LLMs in generating consistent and complete narratives.

**Compliance With Llm Reviewing Policy:**

Affirmed.

**Final Justification:**

I recommend the paper be accepted and my justifications are given above. The authors addressed my questions adequately.

**Key Questions For Authors:**

I actually only have one question.

1) Are you able to further elaborate on the technical difference between facts and commitments? Is it safe to view commitments simply as facts that cannot change throughout the narrative or is there more of a difference than this?

**Limitations:**

yes

**Strengths And Weaknesses:**

Soundness: It is difficult to comment on this criteria since the paper is not making many scientific claims. Rather, it is proposing a benchmark collection of tests one can run to evaluate the effectiveness of an LLM. I suppose the most obvious thing to comment on about "soundness" is the experiments, in which case it seems to be the case that the experiments support the initial concern about LLMs not generating stories well.

Presentation This is a nicely written paper. It cleanly and concisely describes the framework and experimental evaluation details.

Significance: The problem is certainly relevant, but whether this is the "right" approach is unclear. The authors focus narrowly on maintaining facts and commitments and focus on specific reasons a narrative might fail. Of course, to formalize a study one would have to have objective criteria like this. The bottom line is would something that passes this test seem like a good narrative that I would want to engage with during my leisure time? I am not convinced of this. But this work represents a good step in the right direction.

Originality: To be fair, formalizing the process of evaluating LLMs in narrative story generation does seem to be a very difficult task. This work seems to be original in that it is proposing a framework to at least check if LLMs are being coherent (even if not natural).

---

> ### Author Rebuttal · Authors · 2026-03-31
>
> **We sincerely thank the reviewer for the positive assessment and for recognizing the relevance of our work, the originality of our evaluation framework, and the clarity of our presentation.**
>
> ## Technical difference between facts and commitments
> To clarify the distinction: facts represent the active, mutable world state at a given turn, while commitments are predefined narrative constraints that govern the valid progression of these states over time.
> More specifically, facts record objective conditions (e.g., knowledge, injury, captivity) that are updated as the story evolves. Taking the Iron Man as an example, initially, a fact "Tony Stark is uninjured and not yet captured" is true. When a event occurs, the fact is negated, and a new fact reflecting Stark's captivity is added to the active state.
> In contrast, commitments define the logical dependencies that the trajectory must strictly satisfy across multiple turns. For instance, a commitment is an ordering constraint dictating that a event (Stark's critical wounding and capture) must occur before another event (captive surgery). Any storyteller action attempting to trigger captive surgery before the capture event violates this commitment.
> In summary, facts track what is currently true, whereas commitments enforce what narrative logic must be maintained.
>
> ## Focus narrowly on maintaining facts and commitments
> We very much appreciate the reviewer's candid and thought-provoking reflection: "would something that passes this test seem like a good narrative that I would want to engage with during my leisure time?" This is exactly the kind of question we hope our work invites.
> We fully agree that logical consistency is a necessary but not sufficient condition for a truly compelling narrative. A story free of contradictions can still feel flat, predictable, or emotionally disengaged. Our current framework deliberately targets this foundational layer—state coherence and commitment preservation—because our experiments show that even this baseline requirement remains far from solved: the best models succeed in fewer than 14% of cases, and coherence degrades sharply over turns. We believe it is difficult to meaningfully evaluate higher-order narrative qualities (e.g., dramatic tension, emotional resonance, character depth, pacing) when the underlying logical scaffold is itself unreliable.
>
> That said, we see NCP-Bench as a stepping stone toward exactly the richer evaluation the reviewer envisions. Once models can reliably maintain narrative integrity, the natural next questions become: Can they create satisfying dramatic arcs? Can they adapt tone and pacing to the player's engagement? Can they generate genuinely surprising yet coherent plot developments? We are excited about several future directions that build on the current framework:
>
> - Narrative engagement metrics: Incorporating subjective quality dimensions—such as narrative tension, emotional impact, and player agency.
> - Richer commitment structures: Extending commitments beyond ordering constraints to capture thematic arcs, character development trajectories, and dramatic pacing requirements, moving closer to what makes a story genuinely enjoyable.
> - Human-in-the-loop evaluation: Pairing our automated consistency checks with human enjoyment ratings to study the relationship between logical coherence and perceived narrative quality.
>
> We are grateful that the reviewer sees this work as "a good step in the right direction," and we share the ambition of ultimately building systems that produce narratives people would genuinely choose to experience. We hope that by establishing a rigorous and reproducible consistency baseline, NCP-Bench can serve as a shared foundation for the community to pursue these broader goals together.
>
> To further strengthen the paper and address broader feedback across all reviews, we have conducted extensive additional experiments during this rebuttal period. Specifically, we:
> - Evaluated a stronger agentic baseline (HiAgent, ACL 2025), showing that while advanced scaffolding improves survival length, it still fails to solve the core failure modes.
> - Validated evaluator reliability by adding cross-model checks (GPT-5.4-mini and Gemini-2.5-Flash, showing >0.96 correlation) and conducting expert human validation on 100 final results (confirming an very low false-positive rate).
> - Ablated natural vs. adversarial inputs, proving that the benchmark remains fundamentally difficult even with human-like interactions rather than adversarial input.
> - Conducted a human pilot study, demonstrating that the task is structurally feasible for humans, thereby highlighting the significant gap between current LLM capabilities and human performance.
> - Performed an independent expert re-review of the benchmark specifications, verifying its high quality with an average localized issue flag rate of only 2.3%.
>
> **We hope these results resolve your concerns, and thank you again for your support of our work.**

---

> > ### Author Rebuttal · Reviewer_MDPm · 2026-04-05
> >
> > Thank you for addressing my questions, I have no follow up questions that could affect my judgement of the manuscript.

---

> > > ### Author Response · Authors · 2026-04-06
> > >
> > > Thank you for your response and for your supportive evaluation of our paper. We appreciate your thoughtful comments, which helped improve the clarity of our presentation. We are grateful for your time and support.

---

### Official Review · Reviewer_qxmy · 2026-03-13

**Soundness:** 3
**Presentation:** 3
**Significance:** 3
**Originality:** 3
**Overall Recommendation:** 4
**Confidence:** 4

**Summary:**

This paper introduces NCP-bench, a benchmark for evaluating LLMs on commitment preservation, i.e., the capability to preserve logical consistency, in long-horizon interactive narratives. NCP-bench is curated from carefully selected movie synopses and features LLM-based automatic evaluation of conflicts of the generated trajectory with facts, commitments, and user input. Evaluation of several LLMs on NCP-bench shows that even SOTA LLMs struggle to preserve commitment in interactive narrative, highlighting the challenge in maintaining long-horizon logical consistency.

**Compliance With Llm Reviewing Policy:**

Affirmed.

**Final Justification:**

The authors have addressed most of my concerns with the additional experiments, and I have raised the scores.

**Key Questions For Authors:**

Please see the weaknesses above.

**Limitations:**

yes

**Strengths And Weaknesses:**

Strengths
- This paper targets the problem of long-horizon logical consistency preservation, which is critical but overlooked by existing benchmarks. The introduced benchmark provides a potentially useful resource to address this critical challenge.
- The problem formulation and dataset curation are rigorous and well presented.

Weaknesses
- The evaluations are purely automatic and rely on LLM-as-a-judge, there is no evaluation or analysis on the accuracy of the evaluator agents in checking for conflicts.
- The evaluation focuses only on plain LLMs, given that there has been a rich body of work on agentic systems with memories of long-horizon tasks (e.g. [1]), it would be a more comprehensive evaluation to include some of those.

References

[1] Hu et al. HiAgent: Hierarchical Working Memory Management for Solving Long-Horizon Agent Tasks with Large Language Model. ACL 2025

---

> ### Author Rebuttal · Authors · 2026-03-31
>
> **We thank you for acknowledging our problem as critical but overlooked by previous studies, and our problem formulation and dataset curation as rigorous and well presented.
> We agree on the importance of evaluator reliability and stronger long-horizon baselines. We address both concerns with new evidence.**
>
> ## Reliability of the LLM-as-a-Judge evaluator
>
> We agree that evaluator reliability must be validated independently. We therefore added two complementary checks: cross-evaluator sensitivity with an additional GPT-5.4-mini evaluator, and human review of 100 final GPT-4o-mini evaluated results.
>
> Under the same storyteller (GPT-4o-mini), GPT-5.4-mini, Gemini-2.5-Flash, and GPT-5.2 produce highly consistent aggregate outcomes. Based on the three-evaluator summary table, the pairwise Pearson correlations (GPT-5.4-mini / Gemini-2.5-Flash, GPT-5.4-mini / GPT-5.2, Gemini-2.5-Flash / GPT-5.2) are 0.9866, 0.9628, and 0.9857, and the corresponding Spearman correlations are 0.9221, 0.9027, and 0.9817. This indicates that the aggregate conclusions are not tied to a single evaluator backbone.
>
> |Evaluator|Fact conflict|Commit. conflict|User conflict|Success count|Max-turn survival|All-resolved|Trajectory process %|Satisfied process %|
> |:-------:|:-----------:|:--------------:|:-----------:|:-----------:|:---------------:|:----------:|:------------------:|:-----------------:|
> |GPT-5.4-mini|63|26|13|5|3|2|16.64|10.03|
> |Gemini-2.5-Flash|64|32|11|5|5|0|10.57|10.90|
> |GPT-5.2|67|41|15|0|0|0|7.67|11.23|
>
> In addition, we conduct two stages to double checking the conflicts, although our evaluation is automated. During the rebuttal, we also asked human experts to review 100 final results for GPT-4o-mini.
> Across these 100 cases, the union of disagreements contains only 4 cases, and all 4 are fact-conflict false positives. No expert found errors in commitment-conflict or user-input-conflict final outputs. The four disputed cases are all boundary cases of state change or epistemic update. For example, misjudging reasonable space transitions, plot changes, or cognitive updates, or mistaking a character's cognition for objective facts.
>
>
> Taken together, the added evaluator ablation and human review support the same conclusion: the evaluator is not perfect, but its observed errors are rare and concentrated in a narrow class of nuanced fact-transition cases.
>
> ## Missing stronger long-horizon baselines
>
> Thank you for your insightful question.
> We added HiAgent you mentioned as a relevant stronger baseline. To keep the comparison fair, both GPT-4o-mini and HiAgent (built on GPT-4o-mini) are evaluated with the same GPT-5.4-mini evaluator.
>
> Under this unified setting, HiAgent increases average turns from 22.16 to 30.07 and reduces commitment conflicts from 26 to 4, but only slightly reduces fact conflicts from 63 to 60. At the same time, it does not remove the core difficulty: 100-turn survival remains 3, overall successful outcomes decrease from 5 to 3, HiAgent increases user-input conflicts from 13 to 38, and its average satisfied commitment rate is lower (9.15% vs 10.03% for GPT-4o-mini).
>
> |Method|Count|Avg turns|Avg traj.|Avg sati.|Success count|Max turn survival|All resolved|Fact conflict|Commit. conflict|User conflict|
> |:---:|:---:|:---:|:---:|:---:|:---:|:---:|:---:|:---:|:---:|:---:|
> |Baseline|100|22.16|0.17|0.10|5|3|2|63|26|13|
> |HiAgent|100|30.07|0.16|0.09|3|3|0|60|4|38|
>
> These results directly answer the concern. A stronger long-horizon framework can improve some aspects of consistency, especially commitment preservation, but the benchmark remains unsolved. In particular, the gains do not transfer into better overall success, and they come with substantially worse handling of local player inputs. A likely reason is that under memory compression, HiAgent tends to compress concrete player actions into more abstract summaries, which makes it easier to incur user-input conflicts.
>
>
> **In summary, to address your concerns regarding evaluator reliability, we conducted cross-evaluator sensitivity analyses and expert human reviews, demonstrating that our automated evaluation is highly consistent and robust. Additionally, we implemented and evaluated the stronger long-horizon baseline, HiAgent. The new results confirm that while advanced memory architectures can improve certain consistency metrics, our benchmark remains a formidable challenge that is far from being solved.**
>
> **We hope these results resolve your concerns, and will be really appreciated if you reconsider the score of our paper.**

---

> > ### Author Rebuttal · Reviewer_qxmy · 2026-04-04
> >
> > Thanks for the authors' response, it mostly addressed my concerns and I have raised my score accordingly

---

> > > ### Author Response · Authors · 2026-04-04
> > >
> > > Thank you for your response and for reconsidering our paper! We are pleased that our rebuttal addressed your main concerns and appreciate your constructive feedback and your time in reviewing our submission.

---

### Decision · Program_Chairs · 2026-04-30

**Decision:**

Accept (regular)

**Comment:**

Summary: This is a benchmark paper addressing the issue of maintaining logical consistency in (interactive) narratives. They extract environments from movie plots and then test LLMs to see how well they can maintain the logic across such trajectories, finding that the SOTA at the time GPT 5.2 only works about 40% of the time after 20 turns.

Pros:
- The long horizon logical consistency in narrative is a long explored problem in adjacent fields and it is timely to see such a topic being explored in the context of the prevailing paradigm of the times
- The rebuttal strengthened the results by providing (somewhat strong) correlation of their benchmark with human evaluations (given that the the pure LLM as a Judge evals originally are a weakness) and offer a way to "debug" narrative agents

Cons:
- The linear plots are relatively simplistic compared to other forms of narratives (e.g. human written interactive narratives, flashbacks, non-linearities) which contain significantly higher branching factors.
- Logical consistency is one part of what makes an interesting narrative and the writing could use some work situating this with respect to existing literature